# Land Cover Changes in Open-Cast Mining Complexes Based on High-Resolution Remote Sensing Data

**Filipe Silveira Nascimento** [1,2]**, Markus Gastauer** [2]**, Pedro Walfir M. Souza-Filho** [2,3,]*****,
**Wilson R. Nascimento, Jr.** [2]**, Diogo C. Santos** [2] **and Marlene F. Costa** [4]

[1]   Vale S.A. Mina de Águas Claras (MAC), Nova Lima, Minas Gerais 34006-270, Brazil; filipe.silveira@vale.com
[2]   Instituto Tecnológico Vale. Belém, Pará 66055-090, Brazil; markus.gastauer@itv.org (M.G.);
      wilson.nascimento@itv.org (W.R.N.J.); diogo.correa@pq.itv.org (D.C.S.)
[3]   Geosciences Institute, Universidade Federal do Pará. Belém, Pará 66075-110, Brazil
[4]   Vale S.A. Gerência de Meio Ambiente Corredor Norte, São Luís, Maranhão 65085-582, Brazil;
      marlene.costa@vale.com
*****   Correspondence: pedro.martins.souza@itv.org; Tel.: +55-91-3213-5563

**Abstract:** Remote sensing technologies can play a fundamental role in the environmental assessment of open-cast mining and the accurate quantification of mine land rehabilitation efforts. Here, we developed a systematic geographic object-based image analysis (GEOBIA) approach to map the amount of revegetated area and quantify the land use changes in open-cast mines in the Carajás region in the eastern Amazon, Brazil. Based on high-resolution satellite images from 2011 to 2015 from different sensors (GeoEye, WorldView-3 and IKONOS), we quantified forests, *cangas* (natural metalliferous savanna ecosystems), mine land, revegetated areas and water bodies. Based on the GEOBIA approach, threshold values were established to discriminate land cover classes using spectral bands, the normalized difference vegetation index (NDVI), normalized difference water index (NDWI) and a light detection and range sensor (LiDAR) digital terrain model and slope map. The overall accuracy was higher than 90%, and the kappa indices varied between 0.82 and 0.88. During the observation period, the mining complex expanded, which led to the conversion of *canga* and forest vegetation to mine land. At the same time, the amount of revegetated area increased. Thus, we conclude that our approach is capable of providing consistent information regarding land cover changes in mines, with a special focus on the amount of revegetation necessary to fulfill environmental liabilities.

**Keywords:** GEOBIA; *canga* ecosystem; Carajás National Forest; mine land revegetation; satellite images; environmental assessment

## 1. Introduction

The societal and environmental impacts of mining activities have become a great focus of public interest [1]. On the one hand, societies expect the production of inexpensive raw materials to drive the economy. The exploitation of mineral reserves should be embedded in efficient and responsible management plans that have a strong focus on the development of neighboring communities and sustainable land use. This balancing act has led to the emergence of environmental regulations in many countries, including Brazil [2], and a series of self-commitments by the mining industry [3,4].

The transformation of natural ecosystems during open-cast mining reduces the amount of wildlife habitat and endangers populations of rare, endemic or threatened species [5,6]. To reduce these environmental changes, the mitigation hierarchy offers a useful form of guidance [7,8]. According to this framework, measures should be taken to avoid and minimize any potential impacts before

an enterprise starts. Then, international good practices indicate opportunities for remediation, i.e., reverse the residual impacts, and these practices are considered before offsets are outlined to address unavoidable impacts [9]. The four pillars of the mitigation hierarchy process increase environmental sustainability by avoiding net losses or generating positive impacts [10,11] as well as by increasing social acceptance of mining [12].

Remediation includes revegetation, i.e., greening, and the progressive rehabilitation of biodiversity, ecosystem structure and ecosystem function in degraded, damaged or destroyed mine lands [2]. Although considerable progress can be achieved in short time periods [13], such mine land rehabilitation remains challenging, especially in tropical ecosystems, as issues related to species selection, biological invasions, and monitoring the effectiveness of rehabilitation activities are not fully resolved [14]. Uncertain rehabilitation trajectories or unclear outcomes of rehabilitation activities [15] require constant monitoring to effectively regulate mining [16]. In addition, information about the status of environmental rehabilitation is derived from ground data [13]. This monitoring requires accurate quantification of the temporal and spatial extents of revegetation and rehabilitation sites within the mines [17].

An increasing demand for accurate and timely information on the nature and extent of land use and land cover changes in and around open-cast mines highlights the importance of remote sensing methods, which can provide accurate tools to track these changes over time [18–22]. Remote sensing covers large areas and has a higher temporal frequency and lower costs than ground-based investigations. Thus, remote sensing may play a fundamental role in the environmental monitoring of rehabilitation activities in mining areas [1,23] given that geographic information systems, satellite images and digital classification systems are available for the automated detection of land use and land cover changes [24–26]. Specifically, the automated analysis of time series, as proposed by the Landsat-based detection of trends in disturbance and recovery (LandTrendr) approach [27], enables the successful tracking of revegetation dynamics within mines based on vegetation indices [28]. However, because these vegetation indices by themselves are unable to discriminate between natural vegetation and revegetated areas, the separation between classes is based on only historical evidence. Furthermore, the LandTrendr approach is restricted to Landsat imagery with a low spatial resolution (30 × 30 m) and thus is unable to adequately map revegetation activities on steep benches. More recently, high-resolution satellite images have been used to capture land use change dynamics [19,29,30] and mine reclamation patterns [31] via fine-scale land mining interpretation. Hence, we adopt a geographic object-based image analysis (GEOBIA) approach to detect small-scale vegetation suppression and revegetation activities in open-cast mines from high spatial resolution image datasets captured by spaceborne remote sensors.

The objective of this study was to provide a methodology to track land cover and land use changes in mines, including the spatial and temporal dynamics of revegetated mine lands, with a special focus on the recognition of small revegetated areas on steep benches. To do so, we quantified the amount of natural vegetation converted to mining areas as well as the amount of mine land revegetated between 2011 and 2015 in the largest iron ore open-cast mining complex in the world, which is situated in the Carajás National Forest (CNF), in eastern Amazon, Brazil.

## 2. Materials and Methods

### 2.1. Study Site

The study area includes the iron ore open-cast mines from the N4-N5 complex situated in the CNF in the watershed of the Itacaiúnas River, eastern Amazon (Figure 1). The study area has a mean temperature of approximately 25 °C and annual precipitation between 1900 and 2000 mm [32]. The climate of the region is tropical warm, with rainy summers and dry hot winters. The CNF is a protected area dominated by dense and open evergreen forests as well as by semideciduous submontane and montane forest formations that cover the hillsides and lower portions of the landscape. In two mountain

ridges within the CNF, Serra Sul and Serra Norte, patches of banded iron formations outcrop on the mountain tops. These patches are covered by hyperdiverse, endemic savanna vegetation [33,34] that is locally called *canga*. These *cangas* cover the largest iron ore reserves in the world [35] and represent the major business in the region. Since the 1980s, the world's largest iron ore mining complex has been in operation, and it extracts 120 Mt of iron ore annually [36].

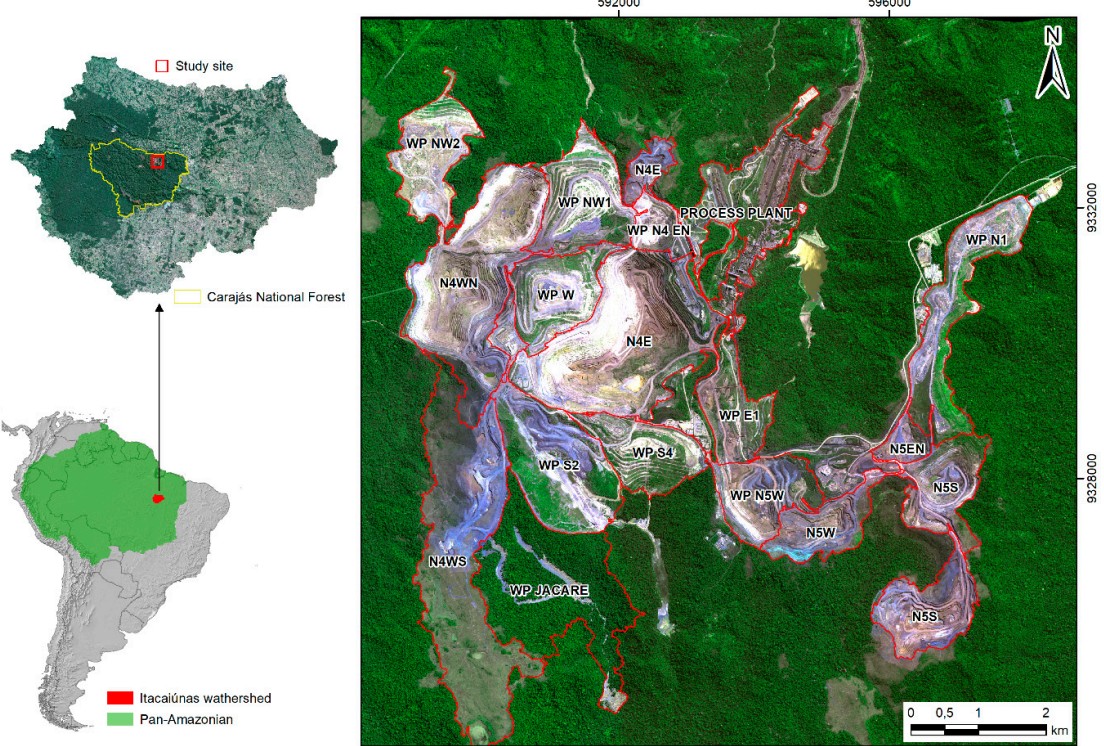

**Figure 1.** Location map of the N4-N5 mining complex indicating mine pits (N) and waste piles (WP) in relation to the Itacaiunas River watershed, southeastern Amazon, Brazil, South America. WorldView-3 satellite image in color-composite 1R2G3B acquired in 2015. The red line represents the boundaries of mining features; WP represents a waste pile; and N represents the mine pits (N is derived from Serra Norte, and the number refers to the original *canga* body, numbered consecutively for each mountain range in the CNF).

During mining operation, the original vegetation, i.e., forests and *cangas*, is logged, and topsoil and mining waste are removed and deposited in waste piles to guarantee access to high-quality iron ore. Benches in active mine pits are revegetated to reduce soil erosion and increase their stabilization. Furthermore, benches from permanent structures, such as waste piles, are subjected to permanent environmental rehabilitation to reduce the overall impact of mining enterprises on natural resources [14]. In both cases, similar mixtures of native and fast-growing and nonnative species are applied together with organic and inorganic fertilizers and fixing material via hydroseeding after bench preparation using biotextiles and/or manual trench digging. The species mixtures used for temporary revegetation in mining pits do not include tree species, as trees may destabilize the landscape by falling; however, the establishment of trees or shrubs from forest and *canga* ecosystems is implemented during the environmental rehabilitation of waste pile benches.

## 2.2. Remote Sensing Data Sources

High-resolution images from three different sensors (2011, 2013 IKONOS, 2012 GeoEye and 2015 WorldView-3) were acquired and used in this research. Table 1 provides detailed information about this remote sensing dataset. All images were acquired in August of each year (i.e., the dry season) to

minimize the cloud cover and the spectral differences in vegetation cover as a function of seasonality along the analyzed time series. The spatial resolution differed among the images, i.e., 4 m for IKONOS, 2 m for GeoEye and 1.24 m for WorldView-3. Additionally, a digital terrain model was used to map the land cover and land use classes in the N4-N5 mining complex. Figure 2 illustrates the main steps of the digital image processing and GEOBIA proposed in this study.

**Table 1.** General characteristics of WorldView-3, GeoEye and IKONOS images acquired during the dry season (August) used in this study.

| Satellite Acquisition date | WorldView-3 1 August 2015 | GeoEye 1 July 2012 | Ikonos 23 May 2011; 22 July 2013 |
|---|---|---|---|
| **Spectral Resolution** | | | |
| Coastal | 400–450 nm | — | — |
| Blue | 450–510 nm | 450–520 nm | 450–520 nm |
| Green | 510–580 nm | 520–600 nm | 520–600 nm |
| Yellow | 585–625 nm | — | — |
| Red | 630–690 nm | 625–695 nm | 630–690 nm |
| Red Edge | 705–745 nm | — | — |
| Near Infrared 1 | 770–895 nm | 760–900 nm | 760–900 nm |
| Near Infrared 2 | 860–1040 nm | | |
| Panchromatic | 450–800 nm | 450–900 nm | 450–900 nm |
| **Spatial Resolution** | | | |
| Panchromatic | 0.3 m | 0.5 m | 1 m |
| Multispectral | 1.24 m | 2 m | 4 m |
| Radiometric Quantification | 11 bits per pixel | 11 bits per pixel | 11 bits per pixel |
| Scene Size | 13.1 km | 15.2 km | 11.3 km |

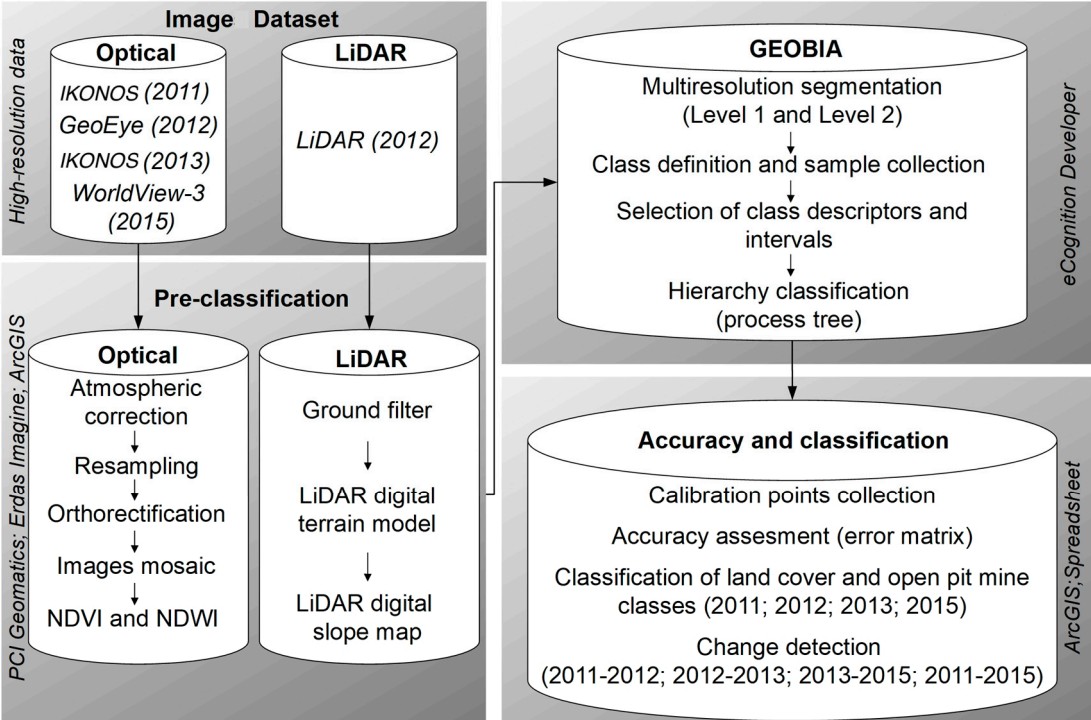

**Figure 2.** Flowchart including the main steps of digital image processing and geographic object-based image analysis (GEOBIA).

### 2.3. Atmospheric Corrections and Orthorectification

Conversions from digital number to ground reflectance data were carried out in the Atmospheric Correction (ATCOR) module of PCI Geomatica 2016 software [37]. Specifically, the ATCOR wizard was applied to calibrate the radiometric models considering the entire scene and to generate the reflectance images. As images may suffer from deformation due to sensor inclination and altitude variations, the images were orthorectified after atmospheric correction. This orthorectification, including georeferencing of the imagery, was carried out in the OrthoEngine Tool of PCI Geomatica [38] using the digital terrain model as the basis for the coordinates (latitude, longitude) and altitude. The root mean square error was approximately one pixel in size. To map the land cover as accurately as possible, the pixel size of GeoEye was 2 m, whereas the IKONOS and WorldView-3 images were resampled to 2 × 2 m during the orthorectification process. Hence, the IKONOS image had the original reflectance preserved, while only WorldView-3 reflectance values needed to be resampled from nearest neighbor interpolation. In the next step, images were mosaicked to cover the complete study site.

### 2.4. Elaboration of Remote Sensing Indices

In addition to the original multispectral bands of the orbital sensors (B1–red, B2–green, B3–blue, and B4–infrared) and the light detection and ranging sensor (LiDAR, see next section for details), a digital terrain model derived from LiDAR was used for image interpretation. We also computed the normalized difference vegetation index–NDVI [39] and normalized difference water index–NDWI [39,40] (Figure 2). The WorldView-3 image presents two spectral bands in the infrared region; we used the spectral band with wavelengths between 770 and 895 nm, which is similar to the near infrared band from further sensors used in this study. The NDVI highlights the presence of vegetation, while the NDWI separates water surfaces from all further land classes. For both indices, higher values indicate the presence of more photosynthetically active vegetation [40].

### 2.5. LiDAR Data Processing

An overflight with a full waveform LiDAR was carried out in June 2012, with six elevation points per square meter. From that, the digital terrain model (DTM) was produced, and ground returns were separated from vegetation returns using the GroundFilter DTM by Triangular Irregular Network interpolation to generate a bare earth surface using the raster calculator in ArcMap. A digital slope map (SM) was created using geoprocessing tools. Both maps were generated in ArcGIS 10.3 software (ESRI) at 2 m spatial resolution.

Using the high-resolution satellite images alone, the *cangas* were often difficult to distinguish from old mining areas (mining areas that remained mining areas throughout the observation period), such as weathered slopes or rocky outcrops within the mine, which presented similar reflectance parameters as the *canga* areas. Hence, the inclusion of LiDAR (DTM and SM) data resulting from a single overflight increased the accuracy of the classification in these cases.

### 2.6. GEOBIA: Image Segmentation, Multilayer Calibration and Hierarchical Classification

Segmentation is the division of a scene into homogeneous parts to extract image objects [41]. The segmentation of the N4-N5 iron ore open-cast mine scenes was carried out by a multiresolution segmentation algorithm based on the homogeneity definition [42]. A four-date segmentation was conducted from all of the ground reflectance bands from 2011 and 2013 IKONOS, 2012 GeoEye and 2015 WorldView-3 with the same resampled pixel size (2 m). To detect land cover and land use changes, we carried out a segmentation process from four separate single-date images, as suggested by Descleé et al. [43] and Duveiller et al. [44]. Multidate segmentation allowed for the comparison of four single images based on objects with the same geometry, delineating spatially and spectrally consistent segments and avoiding misclassification [45]. Hence, it was possible to estimate the most accurate and rapid process, reducing additional processing efforts required to outline polygons [46].

The segmentation approach was developed on two levels using eCognition software (Trimble, München, Germany). First, the macrosegmentation aimed to segregate large features of the image using a less-refined segmentation (scale: 100, compactness: 0.5, and form: 0.1, using weight 1 for all 8 bands). In the second step, a more refined microsegmentation was performed (scale: 30, compactness: 0.5, and form: 0.1, using weight 25 for the NDVI band, 15 for the infrared band, and 1 for the other bands) (Figure 2). Different weights were chosen from a trial and error (heuristics) approach to enhance specific objects that were effectively differentiated in predetermined spectral bands and indices [22,47].

To calibrate the threshold values for the classification of segments, 35 ground control points were used for each class. This procedure was separately carried out for each segmentation step and resulted in the definition of the threshold values shown in Tables 2 and 3. A hierarchical classification was carried out from fuzzy logic [48]. Based on these threshold layer values, macrosegments were classified into forests, *cangas*, mining areas and water bodies using the threshold values shown in Table 2. Objects located in the boundary between *canga* and forest classes (i.e., *canga*-forest transitions) presented different spectral and topographic characteristics. After their identification, these objects were pooled into the *canga* class. The classification of microsegments in the mining area class enables the differentiation between revegetated and non-revegetated mining areas (Table 3). This refinement was necessary to identify revegetated and rehabilitated areas within mines and to separate them from misclassified forests, which were defined as complementary forests 1 and forests 2 and were grouped with the forest class during the first segmentation. Moreover, the DTM and SM derived from LiDAR data were important for discriminating the *canga* class from the mining areas.

**Table 2.** Layers, functions and thresholds used to classify macrosegments in the classes of forests, *cangas*, mining areas and water bodies. NDVI represents the normalized difference vegetation index, NDWI represents the normalized difference water index, DTM represents the digital terrain model and SM represents the digital slope map.

| Class | Layer | Ikonos 2011 | GeoEye 2012 | Ikonos 2013 | WorldView 2015 |
|---|---|---|---|---|---|
| Forests | B1: Red | - | 0–1.9 | - | - |
| | B2: Green | 2.1–5.5 * | 2.1–5.5 * | 2.1–5.5 * | 1.7–38 * |
| | B3: Blue | 2.8–6.3 * | 1.8–6.3 * | 2.5–7.3 * | 0.9–6.3 * |
| | B4: Infrared | - | - | - | - |
| | B5: NDVI | 0.85–1 | 0.76–1 | 0.65–1 | 0.81–1 |
| | B6: NDWI | - | - | - | - |
| | B7: DTM | - | - | - | - |
| | B8: SM | - | - | - | - |
| *Cangas* | B1: Red | 1.3–2.7 | 1.3–2.7 | 1.3–3.4 | 1.3–2.5 |
| | B2: Green | 3.8–5.75 * | 3.8–5.75 * | 3.8–5.75 * | 3.1–5.75 * |
| | B3: Blue | 4.5–8.4 * | 4.5–7.4 * | 4.5–9.3 * | 4.5–7.4 * |
| | B4: Infrared | 18–26 * | 10–18.5 * | 10–18 * | 10–18.5 * |
| | B5: NDVI | 0.78–0.88 | - | - | 0.64–0.8 |
| | B6: NDWI | −0.7−−0.4 | - | - | −0.47−−0.33 |
| | B7: DTM | - | - | - | 562–700 |
| | B8: SM | 0–17.5 * | 0–17.1 * | 0–17.5 * | 0–23 * |
| Complementary *cangas* (threshold condition: objects adjoining *canga* edges) | B1: Red | 0.85–3.3 * | 0.85–3.3 * | 0.85–3.6 * | 0.85–3.3 * |
| | B2: Green | 3–6.5 * | 3–6.5 * | 3–6.5 * | 3–6.5 * |
| | B3: Blue | 3.1–9.6 * | 3.1–9.6 * | 3.1–9.9 * | 3.1–9.6 * |
| | B4: Infrared | - | - | - | - |
| | B5: NDVI | 0.74–0.94 | 0.72–0.94 | 0.57–0.94 | 0.67–0.94 |
| | B6: NDWI | - | - | - | - |
| | B7: DTM | - | 0–700 | 0–700 | 0–722 |
| | B8: SM | - | - | - | - |

**Table 2.** *Cont.*

| Class | Layer | Ikonos 2011 | GeoEye 2012 | Ikonos 2013 | WorldView 2015 |
|---|---|---|---|---|---|
| Mining areas | B1: Red | 0.27–7.7 * | 0.27–9.6 * | 0.27–10 * | 0.27–9.6 * |
| | B2: Green | 1.6–13.5 * | 1.6–14.6 * | 1.6–17 * | 1.6–14.6 * |
| | B3: Blue | 6–25 | 4.3–47 | 4.3–47 | 3.5–47 |
| | B4: Infrared | - | - | - | - |
| | B5: NDVI | - | - | - | - |
| | B6: NDWI | −9.5–0.35 * | −9.5–0.35 * | −9.5–0.35 * | −9.5–0.35 * |
| | B7: DTM | - | - | - | - |
| | B8: SM | - | - | - | - |
| Water bodies | B1: Red | - | - | - | - |
| | B2: Green | - | - | - | - |
| | B3: Blue | - | - | - | - |
| | B4: Infrared | - | - | - | - |
| | B5: NDVI | - | - | - | - |
| | B6: NDWI | 0.1–1 * | 0.1–1 * | 0.1–1 * | 0.1–1 * |
| | B7: DTM | - | - | - | - |
| | B8: SM | - | - | - | - |

* indicates open intervals; otherwise, intervals are closed.

**Table 3.** Layers, functions and thresholds used to classify microsegments from mining areas in the classes of forests, revegetated sites and rehabilitated sites. NDVI represents the normalized difference vegetation index, NDWI represents the normalized difference water index, DTM represents the digital terrain model and SM represents the digital slope map.

| Class | Layers | Ikonos 2011 | GeoEye 2012 | Ikonos 2013 | WorldView 2015 |
|---|---|---|---|---|---|
| Complementary forests 1 | B1: Red | - | - | - | - |
| | B2: Green | 2.1–5.5 * | 2.1–5.5 * | 2.1–5.5 * | 2.1–5.5 * |
| | B3: Blue | 2.8–6 * | 2.8–6 * | 2.8–7 * | 2.8–6 * |
| | B4: Infrared | - | - | - | - |
| | B5: NDVI | 0.87–1 * | 0.87–1 * | 0.73–1 * | 0.87–1 * |
| | B6: NDWI | - | - | - | - |
| | B7: DTM | - | - | - | - |
| | B8: SM | - | - | - | - |
| Revegetated and rehabilitated sites | B1: Red | 1.5–5 * | 1.4–5 * | 1–5 * | 1–5 * |
| | B2: Green | 4.5–11 * | 3.3–11 * | 2–11 * | 2–11 * |
| | B3: Blue | 5.5–14.2 * | 3.6–14.2 * | 2.3–14.2 * | 2.3–14.2 * |
| | B4: Infrared | - | - | - | - |
| | B5: NDVI | 0.7–0.9 | 0.65–0.92 | 0.56–0.92 | 0.6–0.92 |
| | B6: NDWI | −1–−0.3 * | - | - | −1–−0.3 * |
| | B7: DTM | - | - | - | - |
| | B8: SM | - | - | - | - |
| Complementary forests 2 | B1: Red | - | - | - | - |
| | B2: Green | - | - | - | - |
| | B3: Blue | - | - | - | - |
| | B4: Infrared | - | - | - | - |
| | B5: NDVI | 0.78–1 | 0.78–1 | 0.78–1 | 0.78–1 |
| | B6: NDWI | - | - | - | - |
| | B7: DTM | - | - | - | - |
| | B8: SM | - | - | - | - |

* indicates open intervals; otherwise, intervals are closed.

### 2.7. Detection of Land Cover and Open-Cast Mine Changes

In the next step, the polygons were vectorized to calculate the areas of the different classes for each year [22]. To do so, the classifications obtained for each year were exported as shapefiles (.shp) and loaded into ArcGIS 10.3 software (ESRI, Readlands, CA, USA) before the area (in hectares) was calculated using the geometry calculation function present in the layer attributes table. For the analysis

of temporal changes, a conceptual tree of the possible typologies and their transitions was elaborated (Figure 3). Changes between classes during the observation period were detected by comparison among polygons from maps of different dates. Each map was compared to its previous map by subtraction to generate a new thematic map resulting from this differentiation and a table of changes between classes [49]. The "dissolve" geoprocessing function was used for each date of the scene and used to unify polygons from each layer by class. Then, the polygons were compared between subsequent dates using the "Intersect" tool in ArcGIS to produce maps that indicated the land use changes from 2011 to 2012, 2012 to 2013, 2013 to 2015, and 2011 to 2015 (Figure 2).

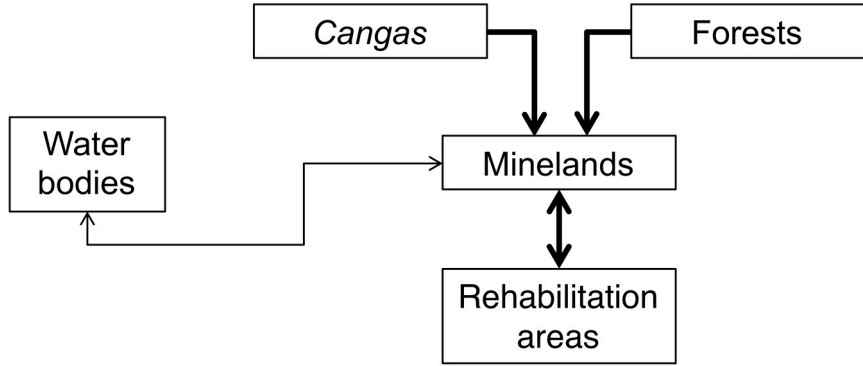

**Figure 3.** Land-cover and land-use classes detected in the N4-N5 mining complex and possible transitions among them. Bold lines indicate most frequent transitions observed in this study.

## 2.8. Classification Accuracy Assessment

To evaluate the accuracy of the GEOBIA classification, polygons were used as the sampling units [50]. It is important to mention that ground control points were not collected in the field. Hence, in each scene, 1024 calibration points were randomly stratified per class in the high-resolution images. This method guaranteed that the number of points was proportional to the size of the land use and land cover class in each scene. For these points, the GEOBIA classification was compared to actual land use and land cover to generate a confusion matrix (Table S1). The actual vegetation was evaluated by manual image analysis by two independent reviewers who were familiar with the scene. Then, the relation of omission and commission errors, the producer's and the user's accuracy, the kappa index per class, the overall kappa index, overall accuracy [50], quantity disagreement (QD) and allocation disagreement (AD) indices [51] were computed for each multitemporal mosaic.

## 2.9. Accuracy Assessment of Land Change

To assess the accuracy and estimate the area of the land change map, 1024 check points were selected to validate the land change map from 2011 to 2015. The objective of this analysis followed best practices to assess the accuracy of the change classification and estimate the area of change in terms of a classification error matrix [52,53]. The error matrix was used to cross-tabulate the land change class labels allocated by the classification of the high-resolution images against the reference check points collected at the sample sites. Accuracy parameters derived from a sample error matrix of unchanged classes (forest, *canga*, mine, revegetation and water) and changed classes (forest to mine, *canga* to mine, mine to revegetation, and revegetation to mine) included overall accuracy, user accuracy of class i (the proportion of the area mapped as class i that was referenced as class i), and producer accuracy of class j (the proportion of the area referenced as class j that was mapped as class j) as proposed by [53].

To assess the accuracy of the land change map, the accuracy was reported in terms of estimated area proportions ($\hat{p}_{ij}$) and not in terms of sample counts, $n_{ij}$. Hence, $p_{ij}$ was substituted by $\hat{p}_{ij}$ and applied for simple random and stratified sampling [53] using the following equation:

$$\hat{p}_{ij} = W_i \frac{n_{ij}}{n_i} \tag{1}$$

where $W_i$ is the proportion of the area mapped as class $i$, $n_i$ is the sum of the mapped class at row $(i, k)$ in the error matrix, and $\hat{p}_{ij}$ is the sum over class $k$. The estimated area of class $k$ is:

$$\hat{A}_k = \hat{p}_k \times A_{total} \tag{2}$$

where $A$ is the total map area. The standard error of the estimated area $S(\hat{A}_k)$ is given by:

$$S(\hat{A}_k) = S(\hat{p}_k) \times A_{total} \tag{3}$$

An approximate 95% confidence interval is obtained as $\hat{A}_k \pm 1.96 \times S(\hat{A}_k)$. For details about the matrix nomenclature, consult Olofsson et al. [53].

## 3. Results

### 3.1. High-Resolution Satellite Image Accuracy Assessment and Estimated Area of Land Change

Based on the evaluation of 1024 points per scene, the overall accuracy of the final land cover and land use maps varied between 90% and 94%; the highest value was obtained in 2011, and the lowest value was obtained in 2012. Therefore, the overall disagreement is very low, and the AD and QD are lower than 8% and 5%, respectively (Tables S1 and S2). The highest kappa index per class was found for forests and *cangas* in all scenes, while the lowest values were detected for water (except in 2011) and rehabilitated areas (Figure 4a). The AD and QD per class were lower than 5.1% and 2.75%, respectively (Figure 4b). The largest user's and producer's accuracy values were found in forest and mining areas (Figure 4c,d). The greatest omission and commission errors of classification were associated with the misclassification of revegetation and water areas (Table 4).

AD allowed for the evaluation of classification as spatial mismatches during map comparisons, thus contributing to the detection of false transitions. QD was important for computing areal differences in classes among maps [22]. Our results showed that the AD of each class was frequently higher than the QD except for the forest and *canga* classes in 2011 and *canga* class in 2013, indicating that the area of the classes tended to be accurate for this purpose. However, the land change detection between different maps tended to be less reliable than that of the computed areas; hence, we expected that none of the results would be significantly affected.

The accuracy assessment and estimated area of land change were calculated from Equation (2) and Equation (3). For example, the estimated area of the mine class was $\hat{A}_{mine} = \hat{p}_{mine} \times A_{total} = 0.184 \times 10,953.7 = 1928.74$ ha. The mapped area of mine ($A_{mine,1}$) was 2019.44 ha; thus, the mine area was overestimated by 90.7 ha. Hence, a degradation area was identified with a 95% confidence interval equal to 2019.4 ± 71.9 ha.

The overall accuracy of the estimated area of land change was 96% (Table 4). The user accuracy varied between 63% (from revegetation to mine) and 99% (unchanged mine), resulting in a slight underestimation of the area changed from revegetation to mine and an overestimation of the unchanged rehabilitation area (Figure 5).

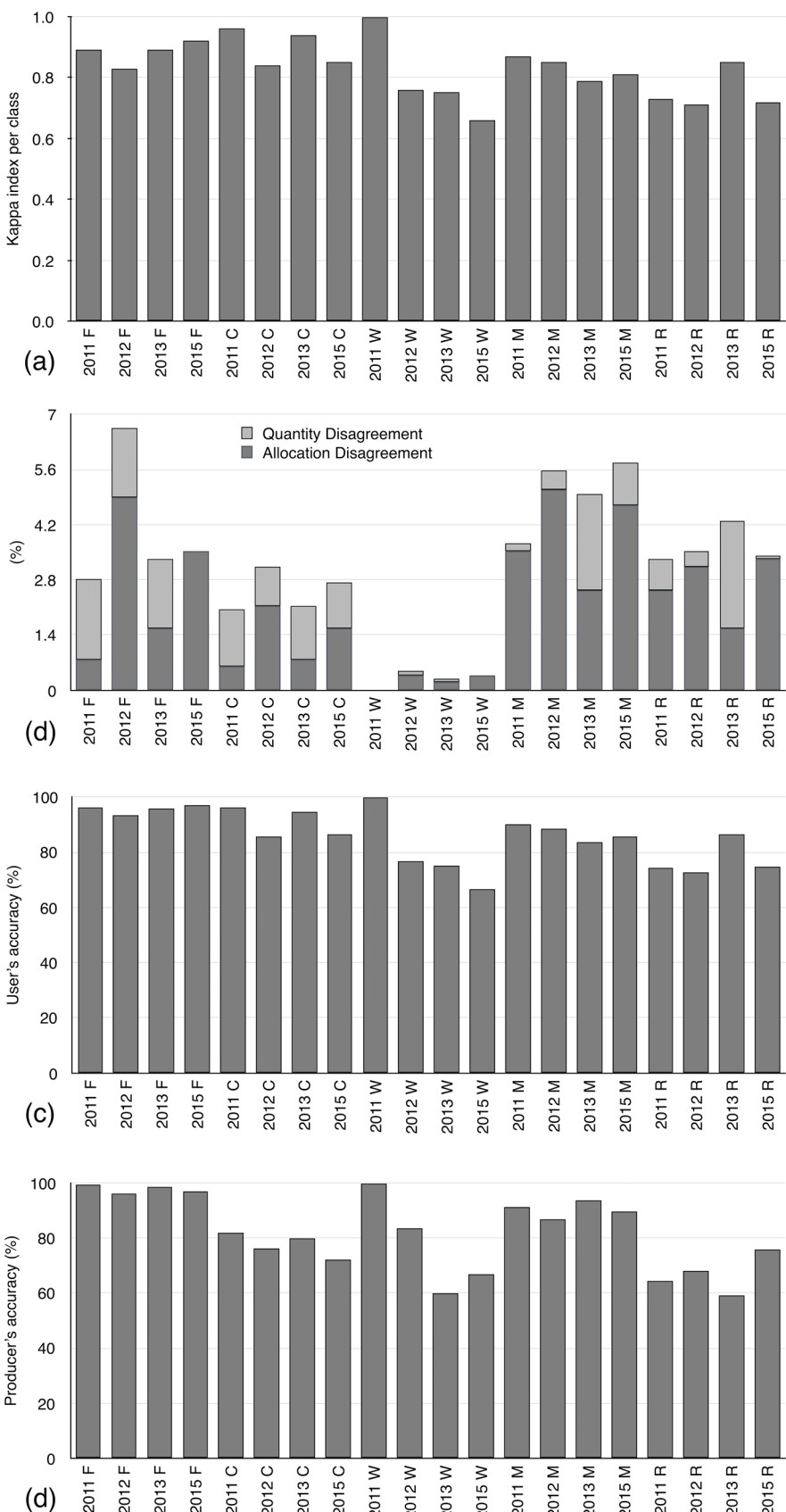

**Figure 4.** Kappa index (**a**), allocation and quantity disagreements (**b**), user accuracy (**c**) and producer accuracy (**d**) per class for 2011, 2012, 2013, and 2015 images in the N4-N5 Carajás mining complex. F is forest, C is *canga*, W is water, M is mine and R is revegetated area.

**Table 4.** Error matrix expressed in terms of classification of the land change map (A) and estimated proportions of areas (B). W = mapped area proportion, UM = unchanged mine, M-REV = from mine to revegetation, C-M = from canga to mine, UC = unchanged canga, F-M = from forest to mine, UF = unchanged forest, UREV = unchanged revegetation, REV-M = from revegetation to mine, ME = Margin of error, and NME = Normalized margin of Error.

| (A) Error Matrix of Classification of the Land Change Map (Line) against the Reference Data (Column) for the Sample Sites | | | | | | | | | | |
|---|---|---|---|---|---|---|---|---|---|---|
| Area | Classes | UM | M-REV | C-M | UC | F-M | UF | UREV | REV-M | Totals |
| 1928.74 | **UM** | 163 | 1 | 0 | 0 | 0 | 0 | 1 | 0 | 165 |
| 377.54 | **M-REV** | 4 | 27 | 0 | 0 | 0 | 0 | 1 | 0 | 32 |
| 353.66 | **C-M** | 1 | 0 | 30 | 0 | 0 | 0 | 0 | 0 | 31 |
| 671.71 | **UC** | 0 | 0 | 3 | 51 | 0 | 1 | 0 | 0 | 55 |
| 264.06 | **F-M** | 0 | 0 | 1 | 0 | 14 | 0 | 1 | 1 | 17 |
| 6790.13 | **UF** | 2 | 0 | 0 | 3 | 0 | 617 | 11 | 0 | 633 |
| 378.08 | **UREV** | 1 | 0 | 0 | 1 | 0 | 5 | 56 | 1 | 64 |
| 189.81 | **REV-M** | 4 | 0 | 0 | 0 | 0 | 2 | 4 | 17 | 27 |
| 10953.7 | **Totals** | 175 | 28 | 34 | 55 | 14 | 625 | 74 | 19 | 1024 |
| | **Producer's accuracy** | 93.1 | 96.4 | 88.2 | 92.7 | 100.0 | 98.7 | 75.7 | 89.5 | |
| | **User's accuracy** | 98.8 | 84.4 | 96.8 | 92.7 | 82.4 | 97.5 | 87.5 | 63.0 | |
| | **Kappa per class** | 0.99 | 0.84 | 0.97 | 0.92 | 0.82 | 0.94 | 0.87 | 0.62 | |
| | **Agreement** | 163.00 | 27.00 | 30.00 | 51.00 | 14.00 | 617.00 | 56.00 | 17.00 | 975.0 |
| | **By chance** | 28.20 | 0.88 | 1.03 | 2.95 | 0.23 | 386.35 | 4.63 | 0.50 | 424.8 |
| | **Overall accuracy =** | 0.952 | **Kappa index =** | 0.918 | | | | | | |
| (B) Error Matrix by Estimated Proportions of Areas | | | | | | | | | | |
| W | Classes | UM | M-REV | C-M | UC | F-M | UF | UREV | REV-M | Totals |
| 0.176 | **UM** | 0.174 | 0.001 | 0.000 | 0.000 | 0.000 | 0.000 | 0.001 | 0.000 | 0.176 |
| 0.034 | **M-REV** | 0.004 | 0.029 | 0.000 | 0.000 | 0.000 | 0.000 | 0.001 | 0.000 | 0.034 |
| 0.032 | **C-M** | 0.001 | 0.000 | 0.031 | 0.000 | 0.000 | 0.000 | 0.000 | 0.000 | 0.032 |
| 0.061 | **UC** | 0.000 | 0.000 | 0.003 | 0.057 | 0.000 | 0.001 | 0.000 | 0.000 | 0.061 |
| 0.024 | **F-M** | 0.000 | 0.000 | 0.001 | 0.000 | 0.020 | 0.000 | 0.001 | 0.001 | 0.024 |
| 0.620 | **UF** | 0.002 | 0.000 | 0.000 | 0.003 | 0.000 | 0.604 | 0.011 | 0.000 | 0.620 |
| 0.035 | **UREV** | 0.001 | 0.000 | 0.000 | 0.001 | 0.000 | 0.003 | 0.030 | 0.001 | 0.035 |
| 0.017 | **REV-M** | 0.003 | 0.000 | 0.000 | 0.000 | 0.000 | 0.001 | 0.003 | 0.011 | 0.017 |
| 1.000 | **Totals** | 0.184 | 0.030 | 0.036 | 0.060 | 0.020 | 0.609 | 0.047 | 0.013 | 1.0 |
| | **Producer's accuracy** | 94.4 | 96.5 | 86.8 | 94.2 | 100.0 | 99.2 | 64.118 | 84.789 | |
| | **User's accuracy** | 98.8 | 84.4 | 96.8 | 92.7 | 82.4 | 97.5 | 87.500 | 62.963 | |
| | **Area (ha)** | 2019.4 | 330.2 | 394.4 | 660.9 | 217.5 | 6674.3 | 515.96 | 140.95 | |
| | **ME (95%)** | 71.9 | 53.4 | 55.5 | 60.2 | 49.3 | 92.0 | 91.6 | 48.0 | |
| | **Area (ha)** | 2019 ± 72 | 330 ± 53 | 394 ± 56 | 661 ± 60 | 217 ± 49 | 6674 ± 92 | 516 ± 92 | 141 ± 48 | |
| | **Overall accuracy =** | 0.96 | | | | | | | | |
| | **Normalized area** | 1.047 | 0.875 | 1.115 | 0.984 | 0.824 | 0.983 | 1.365 | 0.743 | |
| | **NME** | 0.037 | 0.141 | 0.157 | 0.090 | 0.187 | 0.014 | 0.242 | 0.253 | |
| | **Standard error** | 36.663 | 27.255 | 28.325 | 30.700 | 25.166 | 46.957 | 46.753 | 24.481 | |
| | **Standard deviation** | 0.109 | 0.363 | 0.177 | 0.260 | 0.381 | 0.157 | 0.331 | 0.483 | |

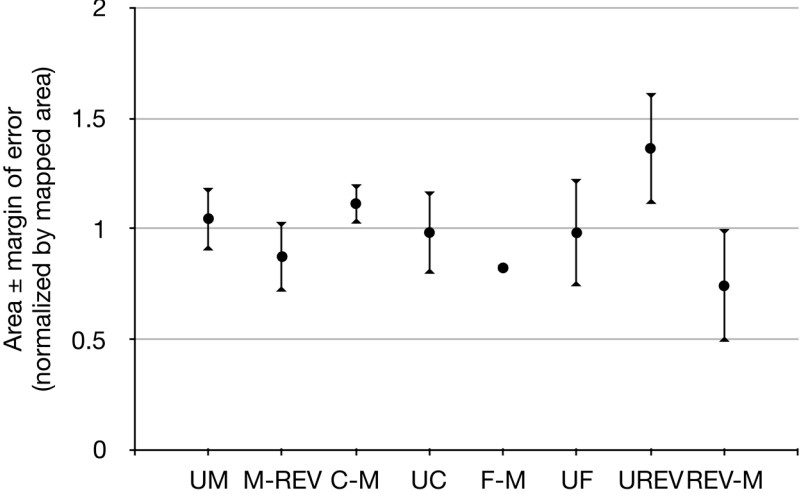

**Figure 5.** Estimated area ± margin of error normalized by mapped area.

### 3.2. Analysis of the Spatial-Temporal Distribution of Land Cover and Land Use Classes

The multiresolution classification with the GEOBIA approach effectively classified the high-resolution satellite images into land cover and open-cast mine classes. Figure 6 illustrates the multitemporal maps throughout the study site for the years 2011, 2012, 2013 and 2015, showing the reduction in forest and *canga* areas and the expansion of mining and revegetated areas.

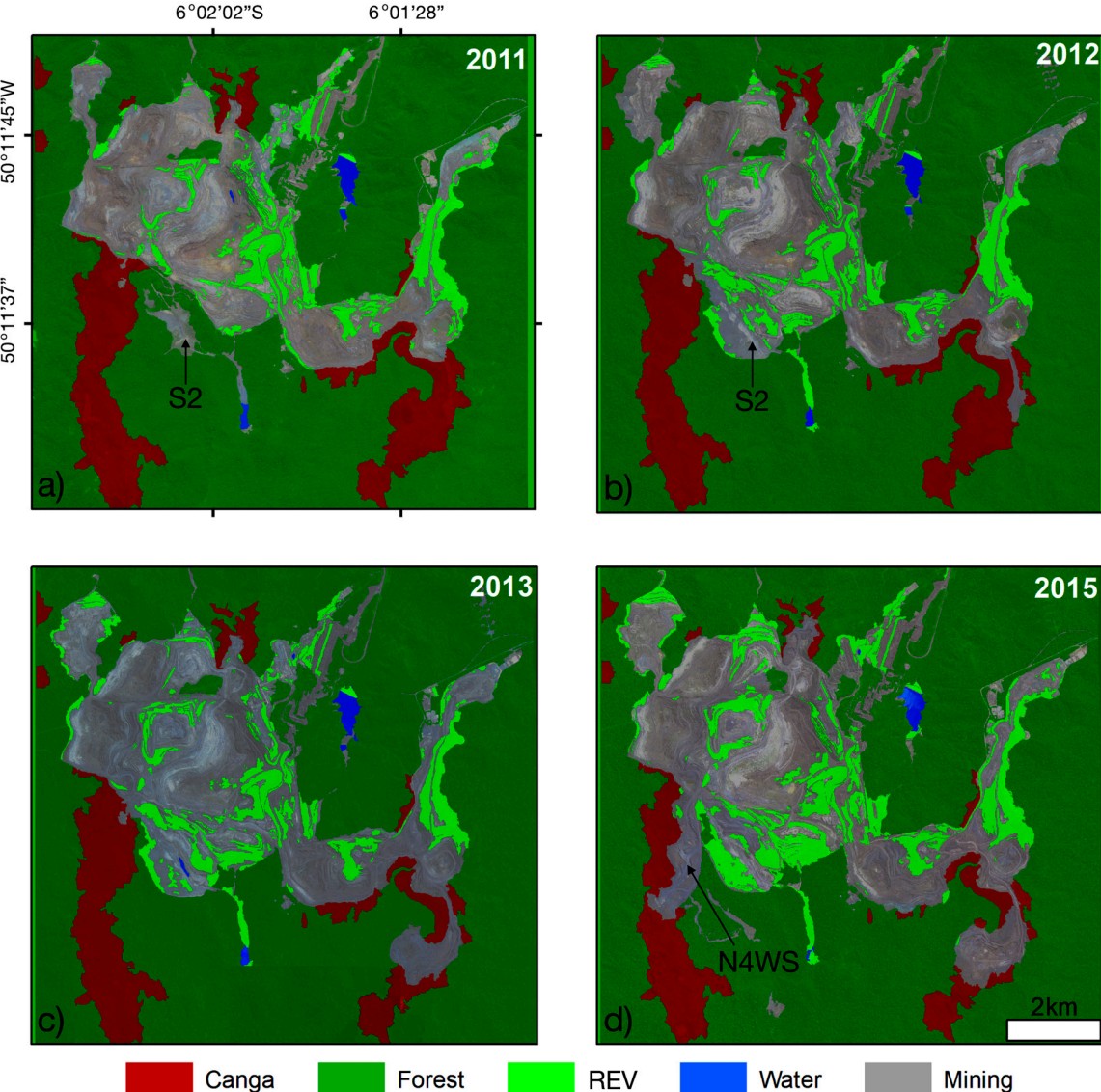

**Figure 6.** Classified high-resolution 2011 Ikonos, 2012 GeoEye, 2013 Ikonos and 2015 WorldView-3 images illustrating the temporal and spatial changes in forest, canga, open-cast mine and revegetated (REV) land cover classes. Figure 6 illustrates the multitemporal maps throughout the study site for the years 2011, 2012, 2013 and 2015, showing the reduction in forest and canga areas and the expansion of mining and revegetated areas.

Between 2011 and 2015, the natural vegetation, i.e., *cangas* and forests, was reduced by 687 ha, while the mining and revegetated areas increased by 420 ha and 279 ha, respectively (Figure 7). Throughout the observation period, forests remained the largest unchanged class in the study site (Figure 8a).

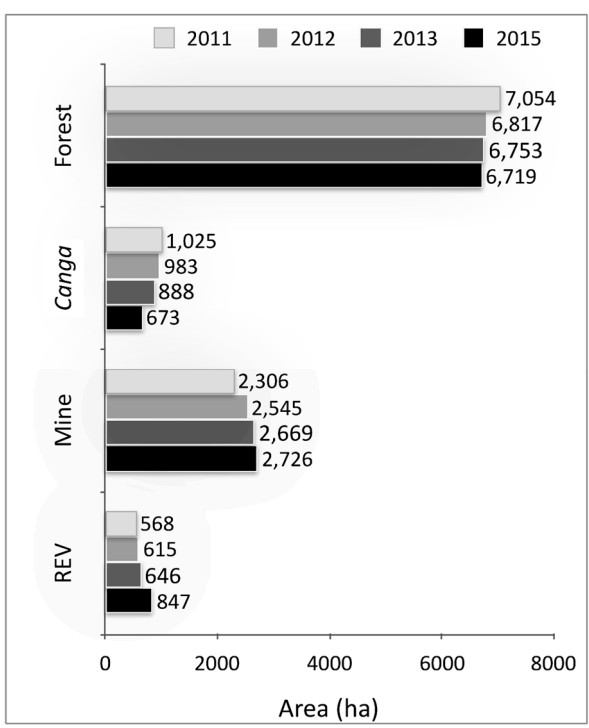

**Figure 7.** Land use classes in the N4-N5 Carajás mining complex between 2011 and 2015. REV indicates revegetated areas.

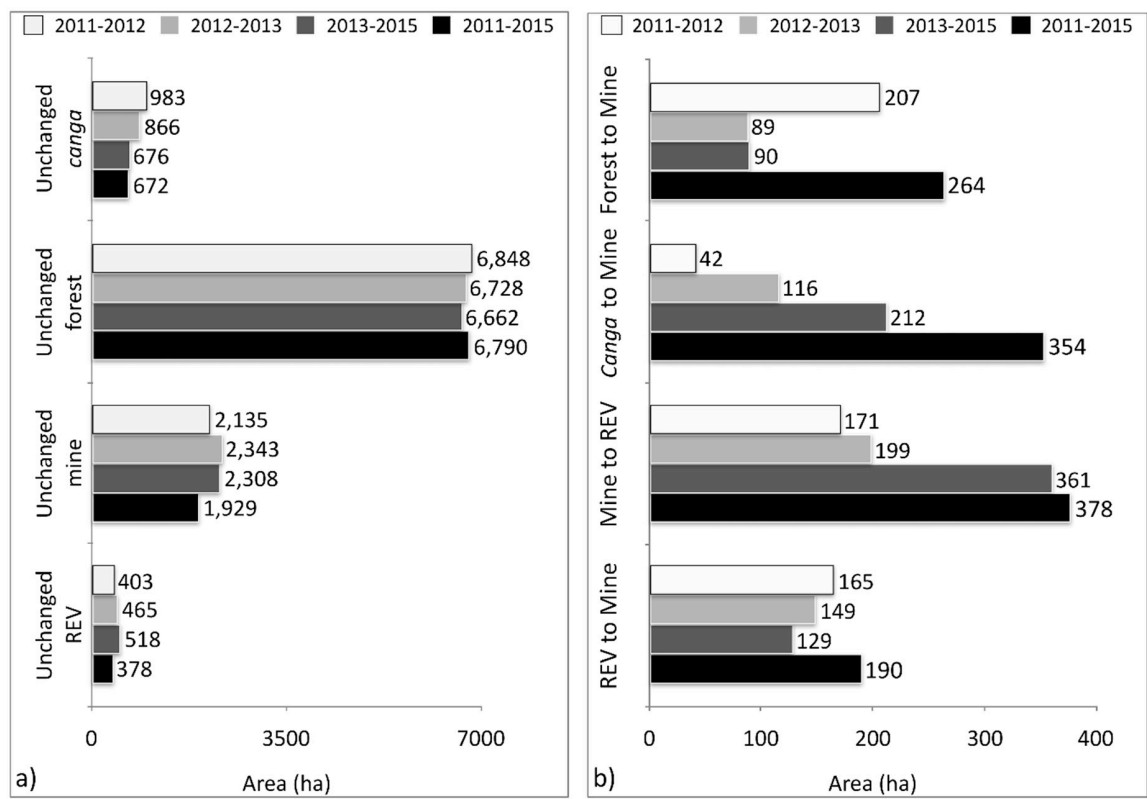

**Figure 8.** Conversion (in ha) of major land cover and land use classes between 2011 and 2015. (**a**) Unchanged *canga,* forest, mine and revegetated areas (REV). (**b**) Conversion of forest to mine, *canga* to mine, mine to REV, and REV to mine.

From 2011 to 2012, unchanged *canga* occupied an area of 983 ha, while unchanged mine and revegetation areas occupied 2135 ha and 403 ha, respectively (Figure 8a). During this period, the largest conversion of forest to mining area that was observed (207 ha) occurred due to the logging of the S2 waste pile area (Figure 8b). At the same time, the area of *canga* converted to mine area was lower than that in the following periods (42 ha). Furthermore, 171 ha of mine land was revegetated; however, 165 ha of this revegetated area was reconverted to mine land (Figure 8b).

In the subsequent observation period (2012 to 2013), the areas of unchanged mine and revegetated land increased by 10% and 15%, respectively (Figure 8b). The conversion of land from forest to mine was reduced by more than 50%, while 116 ha of *canga* was converted to mining area, i.e., an increase of more than 100% compared to the previous period (2011 to 2012). Although 199 ha of mine land was revegetated between 2012 and 2013, the amount of revegetated area increased only marginally because 149 ha of revegetated area was reconverted to mine land (Figure 8b).

Between 2013 and 2015, the unchanged mine area declined in response to the expanding revegetated area, reaching the highest values observed in this study (Figure 8a). This pattern is in response to the establishment of spontaneous vegetation on the S2 waste pile (Figure 1). The conversion of forest to mine increments remained stable (90 ha), while the conversion of canga to mine reached the highest value (212 ha) in response to the installation of the N4 WS mine (Figure 8b). The reuse of revegetated areas by mining activities reached the lowest value of the entire study period (129 ha) (Figure 8b).

This finding indicates a clear tendency of a decrease in *canga* area and its surrounding forest. Throughout the period of change (2011–2015), we observed that the mining area expanded mainly over *canga* (354 ha), followed by forest (264 ha). The revegetation of the mining area increased over time (378 ha), while its reuse by mining decreased (Figure 6). The reformulation of the northern (N) waste pile and the suppression of smaller portions of temporary revegetation in the mining area were responsible for the transformation of revegetated sites in mining areas. Figure 9 illustrates the unchanged land cover and land use (LCLU) classes and the "from-to" change detection classes based on a bitemporal 2011–2015 mosaic image analysis.

Nevertheless, the proportion of revegetated areas in relation to mining areas increased from 24.63% in 2011 to 31.55% in 2015. This increase was partially due to the establishment of spontaneous vegetation at waste pile S2, which, despite being logged, was still not used during the observation period. When we zoomed in on the selected permanent structures, the annual/biannual increases became clearly visible in the following waste piles: Northeast 2–NW2, West–W and South 4–S4 (Figure 10).

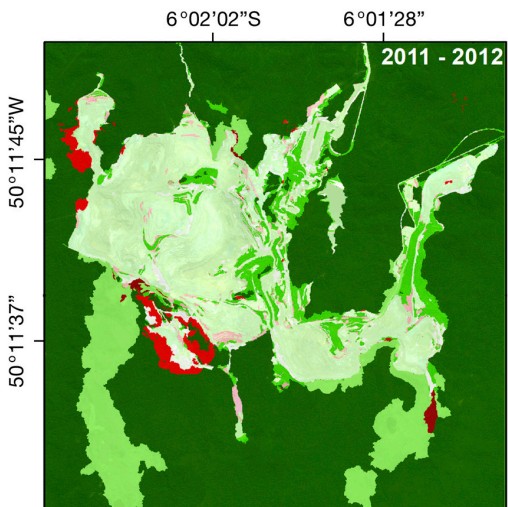 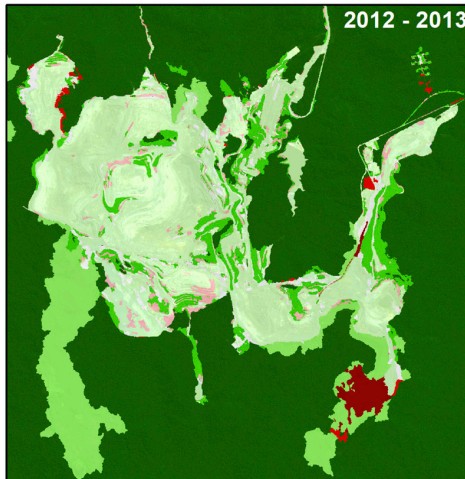

**Figure 9.** *Cont.*

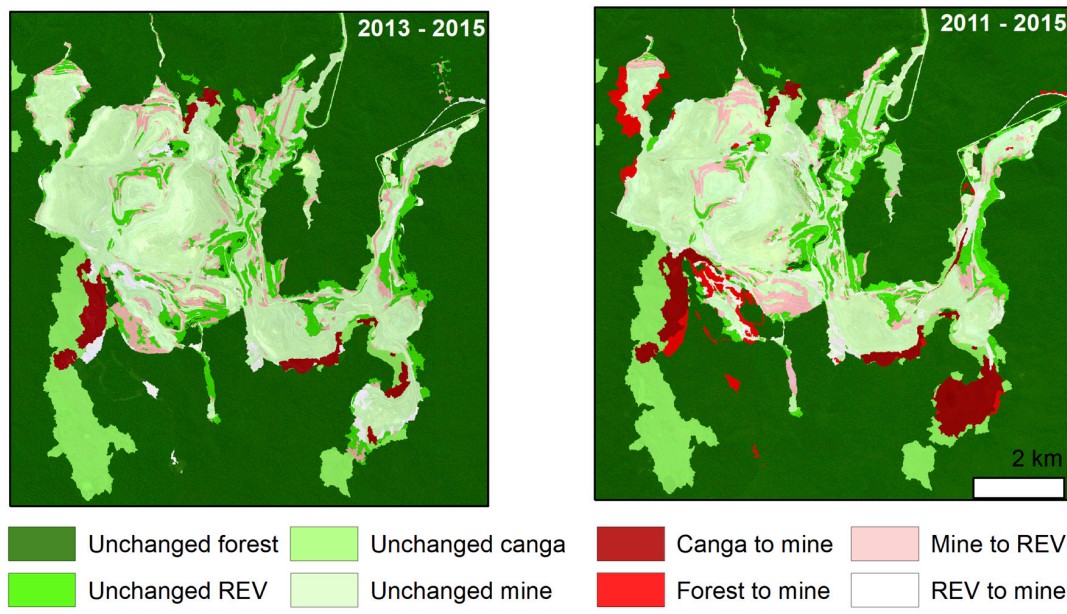

**Figure 9.** Distribution of LULC changes from the GEOBIA during the periods 2011–2012, 2012–2013, 2013–2015, and 2011–2015 using the "from-to" change detection approach.

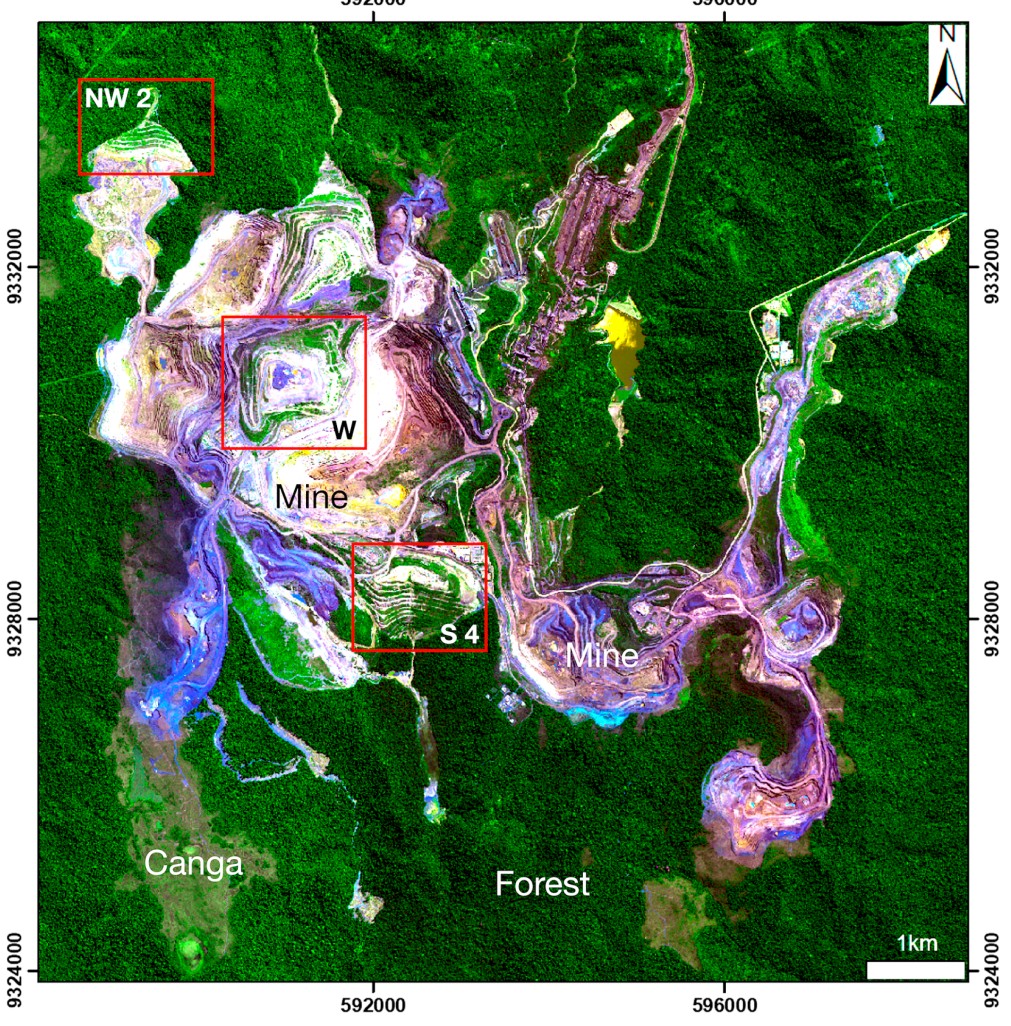

**Figure 10.** *Cont.*

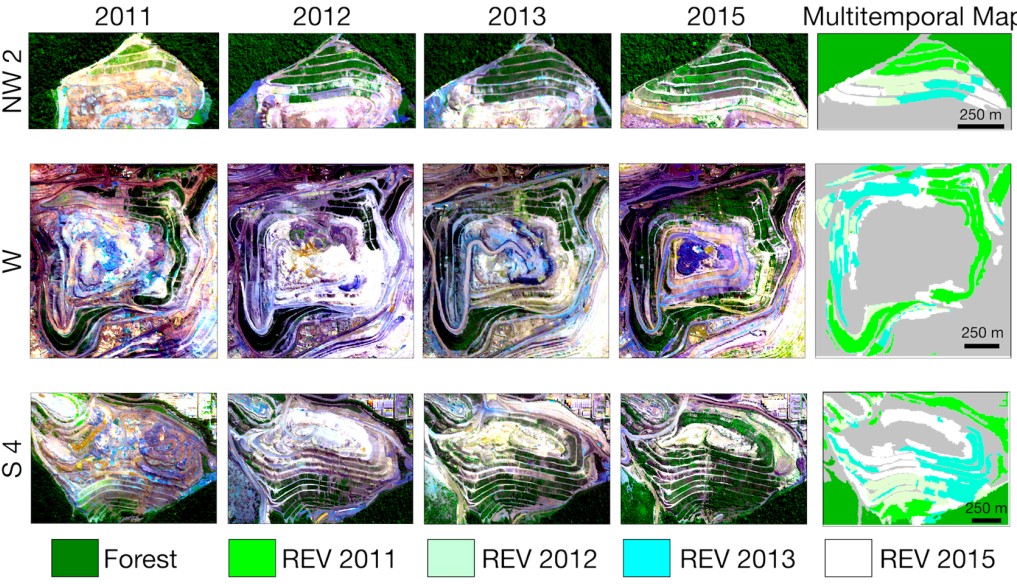

**Figure 10.** Increase in revegetated area (REV) on three selected waste piles (South 4 [S 4], West [W] and Northwest 2 [NW 2]) from the N4–N5 mining complex from Carajás, Brazil, between 2011 and 2015. WorldView 3 satellite image in color-composite 1R2G3B acquired in 2015.

## 4. Discussion

### 4.1. Assessment of the High-Resolution Satellite Image Accuracy and the GEOBIA Approach

The two-step segmentation of the scenes and the utilization of eight different layers, including spectral bands, elevation data and SMs, allowed for the mapping of the spatial land use dynamics of the N4-N5 mining complex from the CNF between 2011 and 2015 with acceptable accuracy. The less detailed scale of the first segmentation step permitted the separation of the entire scene in terms of the areas of natural vegetation, i.e., *canga* and forest vegetation, and mining areas (including mining and revegetated areas). From the level 1 of segmentation, the mining area class was segregated into small-scale (>0.2 ha) revegetated and/or forest areas within the mine. The second segmentation presented a level of detail that was approximately three times greater than that of the first segmentation, allowing the identification of small and medium structures within the mine; this enhanced level of detail enabled us to map the revegetated areas within the mine operations. Thus, despite the use of different sensors that required individual atmospheric corrections and individual calibration, the resolution of our mapping was higher than that obtained from similar approaches [28]. The multispectral classification further allowed the separation of revegetated areas from areas covered by natural vegetation from the time series without preliminary knowledge. The GEOBIA approach presented here showed a straightforward method of mapping the revegetation inside mining areas, and it avoided misclassifications with other vegetation cover that occurred in the neighborhood.

Overall, the kappa indices and accuracies of the classification indicate marginal confusion; these mismatches occurred principally between small revegetated areas and areas of exposed soil in mining sites, and these results confirmed the difficulties in identifying and classifying small features [54]. These limits are set by the spatial resolution of the scenes. However, large revegetated areas from structures such as waste piles were well classified, as the spectral characteristics differentiated them from the surrounding mining areas or natural vegetation. In relation to allocation and quantity disagreement, AD was greater than QD most of the time, indicating that the area of the classes tended to be more accurate for this purpose. However, the change detection between different years tended to be less reliable than the computed areas [22]. The accuracy assessment of land change maps has contributed to estimates of the uncertainty of the area estimates [53], providing more efficient and rigorous methods to estimate the accuracy and area of land change in iron ore mining sites. The major limitation of the

GEOBIA approach was determining an appropriate scale in the two levels of segmentation to improve the classification results. New developments must be made to try to calculate the segmentation scale based on the pixel size and the area of the main target to be mapped.

The individual calibration of scenes originating from different sensors resulted in threshold values that differed slightly between images. Nevertheless, the spectral differences in all scenes between land use classes, including the differences between natural vegetation and revegetated areas, enabled the detection of emerging structures such as areas of vegetation logging, i.e., land conversion from natural ecosystems to mine lands, and emerging rehabilitation sites, i.e., conversion of mine lands to revegetated areas. Thus, these spectral differences were sufficient to track the land use changes and accurately quantify the effectively revegetated areas. However, the assessment of revegetation quality still requires ground monitoring [13]. Specifically, the two-step segmentation of the high-resolution images enabled the identification of small revegetated structures, such as strait slopes, indicating the suitability of the proposed methodology for this study [22].

The evaluation of satellite images by segmentation and classification via eCognition as proposed here represents an effective tool to monitor the spatial extent of revegetation activities within open-cast mines. This technology may be helpful for monitoring the revegetation of mine lands conducted by mining companies and thus could become an important tool for environmental monitoring by official licensing agencies. Furthermore, the approach proposed here is suitable for monitoring revegetation activities; the intersection and quantification of revegetated areas with areas planted by operations contributes to the identification of revegetation constraints, e.g., inadequate planting periods or substrates disabling plant survival, thus contributing to more effective mine land revegetation.

The reduction in natural vegetation and the increase in mining area show that the N4–N5 Carajás mining complex is a fully expanding mine and indicates that the iron ore reserves are not yet exhausted. The amount of expansion (absolute as well as relative) of revegetated areas within the mining complex shows that the mine land has been revegetated, which is the first step to address environmental liabilities by mining companies. Ground points are necessary to assess the quality of the revegetation, i.e., to assess whether revegetated areas are resilient ecosystems as required by legislation [14]. The combination of ground investigations with additional remote investigations of the scene may result in approaches that can be used to map the vegetation structure [55] or species composition of revegetated areas, which may enable the tracking of the advance of revegetated sites towards rehabilitated ecosystems [56].

### 4.2. Revegetation Analysis from GEOBIA using High-Resolution Satellite Data

Revegetation activities in the N4-N5 mining complex are carried out for the temporary revegetation of benches to protect them from soil erosion and reduce dust loading during periods in which they are not in use. Furthermore, permanent revegetation is carried out on permanent structures, such as waste piles, to rehabilitate the biodiversity and ecosystem functioning of mine lands, reducing the overall impact of mining operations. Our classification was unable to differentiate between temporary and permanent rehabilitation areas, which is not surprising because similar seed mixtures and rehabilitation techniques, such as hydroseeding, biotextiles or manual trench digging, were applied for both purposes. Nevertheless, the high amount of revegetated area that was transformed to mining area (i.e., one-third of all revegetated areas mapped in 2011) indicates that a high amount of mine land greening is composed of temporary revegetation. In contrast, approximately two-thirds of the revegetated area mapped in 2011 may correspond to permanent revegetation, and this land is intended to rehabilitate mine land to improve long-term biodiversity and ecosystem services and reduce the overall impact of mining operations.

Our classification revealed that only a very small amount (0.42 ha) of revegetated area was confounded with natural forest vegetation during the observation period (less than 0.05% of the entire rehabilitated area). However, this figure shows that the revegetated mining structures maintained their spectral profiles during the observation period, which is expected because environmental rehabilitation

in tropical ecosystems may span centuries [57], while the studied revegetated areas are, at most, decades old.

Assessing the surface mining area through high-resolution satellite images has a high potential to become an operational tool to monitor and evaluate the dynamics of land cover and land use changes in open-cast mining complexes. It is important to emphasize that high spatial resolution satellite imageries have been commercially available since 1999 with the IKONOS system. Therefore, high-resolution satellites at one meter and below have been operating for more than twenty years, thus providing a substantial amount of spatial and spectral information that is extremely useful for the recognition of the geomorphic features of mines [58] and analysis of different trajectory types of land cover and land use in mining areas [30]. Thus, the GEOBIA approach presented in this paper can be used to study other sites of open-cast mines. Enabling the accuracy assessment and the estimation of the area of land change can become a good practice for monitoring these complex man-made environments. Further methodological advances should focus on the recognition and discrimination of different stages of rehabilitation in mining areas (e.g., herbaceous, shrub and forest cover) from high-resolution satellite and unmanned aerial vehicle systems to remotely track environmental advances of revegetated areas.

## 5. Conclusions

Our classification permitted the mapping of land cover and open-cast mine changes in the N4-N5 mining complex with sufficient accuracy. After the atmospheric correction of images, the generation of the digital terrain and slope models, and calibration of the threshold values for individual scenes, the proposed methodology was shown to be sufficiently robust to monitor land use changes in mining sites, offering a powerful tool for all stakeholders. Specifically, the remote sensing of mine land revegetation dynamics was able to monitor the occurrence of revegetation activities and the attendance of the mitigation hierarchy as required by national environmental laws. It is important to highlight the importance of the proposed two-step segmentation and multispectral classification to accurately separate the land cover and land use classes in the mining scene. Combined with ground data, additional remote investigations of the scene should be used to develop classification approaches for revegetated sites according to their structure or species composition, which may be useful for tracking advances in rehabilitating sites towards natural ecosystems. Moreover, the use of multitemporal high spatial resolution satellite images can become an operational tool for the monitoring of cumulative changes in land use related to mining operations.

**Supplementary Materials:** The following are available online at http://www.mdpi.com/2072-4292/12/4/611/s1, Table S1. Confusion matrix for image classification: (a) 2011 Ikonos; (b) 2012 GeoEye; (c) 2013 Ikonos; and (d) 2015 WorldView-3. The matrix shows the number of verification points, omission (OE) and commission (CE) errors, user (UA) and producer's (PA) accuracy, Kappa index per class, overall accuracy (OA). Table S2. Normalized confusion matrices: (a) 2011 Ikonos; (b) 2012 GeoEye; (c) 2013 Ikonos; and (d) 2015 WorldView-3. The matrix shows the number of verification points, omission (OE) and commission (CE) errors, user (UA) and producer's (PA) accuracy, Kappa index per class, overall accuracy (OA), overall disagreement (OD); allocation disagreement (AD); and quantity disagreement (QD).

**Author Contributions:** Conceptualization, P.W.M.S.-F.; field work collection, F.S.N., W.R.N.J., D.C.S., P.W.M.S.-F., M.F.C.; digital image processing, GIS and formal analysis, accuracy assessments, F.S.N., W.R.N.J., D.C.S., P.W.M.S.-F., writing–original draft preparation, M.G., F.S.N., P.W.M.S.-F., review and editing, P.W.M.S.-F., M.G. All authors have read and agreed to the published version of the manuscript.

**Funding:** This research received no external funding.

**Acknowledgments:** The authors thank the Vale Institute of Technology (ITV) for its management support. The authors are also grateful to the Vale Mining Company for logistical support. PWMSF was supported by CNPq through research scholarships. This project was carried out in the Carajás National Forest with permission from IBAMA (SISBIO 35594-2).

**Conflicts of Interest:** The authors declare no conflicts of interest.

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
