# Peer review of "Land Cover Changes in Open-Cast Mining Complexes Based on High-Resolution Remote Sensing Data"

_remotesensing, doi:10.3390/rs12040611_

Round 1
Reviewer 1 Report
I think that the authors have done a good job in making some significant improvements in the paper and addressing a number of my comments. However, I have 2 outstanding issues that are related to methodology:
The resampling of 4m imagery to 2m is not standard or appropriate practice. If 4m imagery represents a mix of reflectance values of vegetation and exposed soil from mining, then what do the resampling pixels create? Typically is ok to resample to a courser resolution, but not a finer resolution. I would still suggest that PIF normalisation is still appropriate given different sensors and different years of sampling.I think that the paper is worthy of publication and if the authors want to address these issues further, then i am happy for it to be published. Regarding the resampling, the authors could consider either resampling everything to 4m and rerunning the experiment, or drop the IKONOS imagery from the analysis and just use the 2 datasets from WordView-3 and Geoeye.
Other comments:
Line 143 –you have repeated yourself here
Line 154 –add that this is derived from LiDAR
Table 1 – can you add date of capture?
Figure 2 – add the resampling step to the flow chart
Line 247-note that DSM usually is an acronym used for digital surface model. I would consider using SM instead, which you have used in your response to my comments in the first round.
Figure 3 is much improved. However are you suggesting that the cangas and Forests can transition directly into rehabilitation? This seems a little odd. Is this Canga/Forest that is disturbed, but unmined? You don’t have a class for this in the paper? Consider taking those arrows out of the figure.
Lines 460-463 –it is good that you are discussing limitations, but you need more than this. Scale is a massive limitation here –what features are you missing b/c of the scale? The issue of mixed reflectance here should be discussed, particularly b/c you are resampling some imagery to courser resolution and some imagery to coarser resolution. Also different sensors, using different band widths etc
Line 471. Glad you have added this.
Line 486 –I don’t think it is appropriate to speculate on the positive social and economic outcomes from mining here. I think you should keep the discussion to the science you have presented.
Line 488 – You have mentioned env liabilities here and in the in the abstract. I’m not sure it is appropriate to do this unless you want to be more specific about the particular liabilities are they are addressing. My previous review talked about how this paper is not providing evidence of this – the data shows some rehab is occurring but you cannot make broad assumptions that the company is meeting its environmental requirements based on this data. You could say that the results suggest that the mine is revegetating the site and may therefore meet one of its key requirements which is to rehabilitate the site following mining. But of course there are other issues such as erosion, landform stability, subsidence, water, dust etc which need on ground investigation before you can make a statement such as “the env liabilities are being addressed by the company’. Such a sentence is misleading.
Line 533 do you mean shrubs or scrubs?
Line 546 –consider replacing the word ‘efficiency’ with ‘occurrence’ Efficiency implies a quality of rehab
Author Response
Answer to reviewer
I think that the authors have done a good job in making some significant improvements in the paper and addressing a number of my comments. However, I have 2 outstanding issues that are related to methodology:
The resampling of 4m imagery to 2m is not standard or appropriate practice. If 4m imagery represents a mix of reflectance values of vegetation and exposed soil from mining, then what do the resampling pixels create? Typically is ok to resample to a courser resolution, but not a finer resolution. I would still suggest that PIF normalisation is still appropriate given different sensors and different years of sampling.
Answer: I understand your point of view. For long time I thought this way. However, the resampling of 4m imagery to 2 m is done only to all images has the same pixel size. Hence, when we carry out image segmentation, the image set has the pixel size and the original reflectance per pixel. If I resample the better resolution images to 4m pixel size, I will change the original reflectance data, transforming 4 pixels (2x2m) with different reflectance value in just 1 pixel (4x4m) with new reflectance value. Due to explanation, I rather to break one pixel in four pixel with same value. Hence, only WorldView-3 reflectance values needed to be resampled.
I think that the paper is worthy of publication and if the authors want to address these issues further, then i am happy for it to be published. Regarding the resampling, the authors could consider either resampling everything to 4m and rerunning the experiment, or drop the IKONOS imagery from the analysis and just use the 2 datasets from WordView-3 and Geoeye.
Answer: I disagree. See explanation above. A new sentence was inserted in the section 2.3: …the Ikonos and WorldView-3 images were resampled to 2x2 m during the orthorectification process. Hence, the Ikonos image had he original reflectance preserved, while only Hence, only WorldView-3 reflectance values needed to be resampled.
Other comments:
Line 143 –you have repeated yourself here
Answer: We changed the text.
Line 154 –add that this is derived from LiDAR
Answer: We inserted in the text.
Table 1 – can you add date of capture?
Answer:
Figure 2 – add the resampling step to the flow chart
Answer: It is not necessary. The resampling was carried out during the orthorectification process that is explained in the text (section 2.3.).
Line 247-note that DSM usually is an acronym used for digital surface model. I would consider using SM instead, which you have used in your response to my comments in the first round.
Answer: We change DSM to SM along the text.
Figure 3 is much improved. However are you suggesting that the cangas and Forests can transition directly into rehabilitation? This seems a little odd. Is this Canga/Forest that is disturbed, but unmined? You don’t have a class for this in the paper? Consider taking those arrows out of the figure.
Answer: We follow your suggestion.
Lines 460-463 –it is good that you are discussing limitations, but you need more than this. Scale is a massive limitation here –what features are you missing b/c of the scale?
Answer: In this paper, none. But it is impossible, for example, to map different vegetation species in rehabilitating areas with 2 m in spatial resolution.
The issue of mixed reflectance here should be discussed, particularly b/c you are resampling some imagery to courser resolution and some imagery to coarser resolution. Also different sensors, using different band widths etc
Answer: We think that many of these issues are already discussed in the manuscript (see bellow). The manuscript is too long with more than 9,000 words. I am not sure if readers will keep their focus in a so long discussion.
“The individual calibration of scenes originating from different sensors resulted in threshold values that differed slightly between images. Nevertheless, the spectral differences in all scenes between land use classes, including the differences between natural vegetation and revegetated areas, enabled the detection of emerging structures such as areas of vegetation logging, i.e., land conversion from natural ecosystems to mine lands, and emerging rehabilitation sites, i.e., conversion of mine lands to revegetated areas. Thus, these spectral differences were sufficient to track the land use changes and accurately quantify the effectively revegetated areas. However, the assessment of revegetation quality still requires ground monitoring [13]. Specifically, the two-step segmentation of the high-resolution images enabled the identification of small revegetated structures, such as strait slopes, indicating the suitability of the proposed methodology for this study [22].”.
Line 471. Glad you have added this.
Answer: Thanks
Line 486 –I don’t think it is appropriate to speculate on the positive social and economic outcomes from mining here. I think you should keep the discussion to the science you have presented.
Answer: We removed the sentence. “Additional canga blocks and forest portions are expected to be logged in the near future to provide high-quality ore to satisfy international markets and generate economic and social welfare for the region.”
Line 488 – You have mentioned env liabilities here and in the in the abstract. I’m not sure it is appropriate to do this unless you want to be more specific about the particular liabilities are they are addressing. My previous review talked about how this paper is not providing evidence of this – the data shows some rehab is occurring but you cannot make broad assumptions that the company is meeting its environmental requirements based on this data. You could say that the results suggest that the mine is revegetating the site and may therefore meet one of its key requirements which is to rehabilitate the site following mining. But of course there are other issues such as erosion, landform stability, subsidence, water, dust etc which need on ground investigation before you can make a statement such as “the env liabilities are being addressed by the company’. Such a sentence is misleading.
Answer: we agree with you that our data are not sufficient to conclude that environmental liability are addressed. To make this point clear, we revised the section.
Line 533 do you mean shrubs or scrubs?
Answer: We change scrubs to shrubs
Line 546 –consider replacing the word ‘efficiency’ with ‘occurrence’ Efficiency implies a quality of rehab
Answer: We follow your suggestion
Reviewer 2 Report
The article "Land cover changes in open-cast mining complexes based on high-resolution remote sensing data" by Nascimento et al. describes a case study of object-based land use land cover change classification in an open-pit mining area using high-resolution remote sensing images from GeoEye, Ikonos, and WorldView-3 satellite images. While the case study might provide some insights into a classification processing chain especially using high-resolution satellite images, I find a number of major issues. I am sorry that I, therefore, recommend rejecting the paper.
Major 1:
Applying a change detection involves a time1 classification and time2 classification and a combination to obtain the change in between. It seems that the lidar data in-between time 1 and 2 (or3) is taken for each of the individual classifications. That seems to me like a fundamental flaw. Why do the authors use a dataset that is obtained in between for both classifications and how should a change be considered by this method?
Major2:
Why is all the text crossed out? It is very hard to read, I would suggest providing a final revised version for peer review. To show changes in the text, a version such as the one presented here should be additionally provided, if at all.
Major3:
Figure 4 is so hard to read. What is your X Axis ? Time? Why start again? I think I know what the authors are trying to tell with this figure, but I would suggest a careful redesign.
Major 4:
Figure 3 Is this a result or methods? How can forests become rehab areas? The other changes are obvious. I do not see the sense of this figure.
Major5:
I do not understand Figure 5, what is the error bar?
Major6:
The discussion needs a careful investigation and more structure, the authors present two and a half pages of straight text, which has the potential to frighten away potential readers.
Minor1:
What band combination is 1R2G3B? Most Software Packages work with RGB. Doesn't make sense! (Figure 1 )
Minor2:
Table 1
The radiometric resolution is wrong here, you mean radiometric quantification. See basic remote sensing literature.
Minor3:
291-293: That is obvious, please shorten or better leave out.
Minor4:
The Kappa Index is not used any more, I think overall accuracy is enough, but I do not really care.
Minor5:
318 8 and 5 what?
Minor 6:
Under classes in the error matrix there is so much space, yet the authors use all those abbreviations why? Makes this figure nearly unreadable.
Author Response
Answer to reviewer
The article "Land cover changes in open-cast mining complexes based on high-resolution remote sensing data" by Nascimento et al. describes a case study of object-based land use land cover change classification in an open-pit mining area using high-resolution remote sensing images from GeoEye, Ikonos, and WorldView-3 satellite images. While the case study might provide some insights into a classification processing chain especially using high-resolution satellite images, I find a number of major issues. I am sorry that I, therefore, recommend rejecting the paper.
Major 1:
Applying a change detection involves a time1 classification and time2 classification and a combination to obtain the change in between. It seems that the lidar data in-between time 1 and 2 (or3) is taken for each of the individual classifications. That seems to me like a fundamental flaw. Why do the authors use a dataset that is obtained in between for both classifications and how should a change be considered by this method?
Answer: We used the digital terrain model (DTM) and slope model (SM) only to discriminate canga feature. The DTM ranged from 0-700m and SM 0-23o. If occur changes in the relief due to mining activities, the canga area was suppressed and altitude decreased bellow 700m and slope was steeper. Hence, it was possible use the LiDAR data to better discrimination of canga area.
Major2:
Why is all the text crossed out? It is very hard to read, I would suggest providing a final revised version for peer review. To show changes in the text, a version such as the one presented here should be additionally provided, if at all.
Answer: As the journal guideline, I could only upload the revised document with track changes. In other words, without the changes in the word revised document. Unfortunately, I have no control about this process.
Major3:
Figure 4 is so hard to read. What is your X Axis ? Time? Why start again? I think I know what the authors are trying to tell with this figure, but I would suggest a careful redesign.
Answer: X Axis presents the year followed by class according figure caption (F is forest, C is canga). This is a new format to express accuracy assessment result in graphical form.
Major 4:
Figure 3 Is this a result or methods? How can forests become rehab areas? The other changes are obvious. I do not see the sense of this figure.
Answer: It is a method. It is a conceptual model to define and indicate the land use changes used in the from-to change detection approach.
Major5:
I do not understand Figure 5, what is the error bar?
Answer: It is the margin of error normalized by mapped area according Olofsson’s proposal.
Major6:
The discussion needs a careful investigation and more structure, the authors present two and a half pages of straight text, which has the potential to frighten away potential readers.
Answer: We subdivided discussion section in two subsessions: 4.1. Assessing high-resolution satellite image accuracy assessment and GEOBIA approach and 4.2. Revegetation analysis from GEOBIA using high-resolution satellite data
Minor1:
What band combination is 1R2G3B? Most Software Packages work with RGB. Doesn't make sense! (Figure 1 )
Answer: The band combination is 1R2G3B. I do not understand your comment about “does not make sense”.
Minor2:
Table 1
The radiometric resolution is wrong here, you mean radiometric quantification. See basic remote sensing literature.
Answer: I follow your suggestion.
Minor3:
291-293: That is obvious, please shorten or better leave out.
Answer: I rather to keep the original text.
Minor4:
The Kappa Index is not used any more, I think overall accuracy is enough, but I do not really care.
Answer: I rather to keep the original text.
Minor5:
318 8 and 5 what?
Answer: We corrected the text. Allocation disagreement and quantity disagreement are lower than 8% and 5%, respectively.
Minor 6:
Under classes in the error matrix there is so much space, yet the authors use all those abbreviations why? Makes this figure nearly unreadable.
Answer: I rather to keep the original text.
Round 2
Reviewer 1 Report
The authors have attempted to address the resampling methods, but the description in section 2.3 is inadequate. More information is required. For example, I understand that the 4m IKONOS imagery was resampled to 2m without changing the reflectance values. What about WV-3 imagery? What resampling process did this use? Were pixel values averaged? Also, despite the authors refusal, I believe that the resampling should be added to the flowchart to make this process clear.
The authors also have still not addressed the request to use PIF normalisation despite this being raised on numerous occasions.
Limitations need to be added to the discussion and these should at least discuss improvements to methods which could be applied. This is important since the authors are citing the methods used in this paper as appropriate for determining land use changes in mining, however the methods used in the paper could be vastly improved.
Author Response
The authors have attempted to address the resampling methods, but the description in section 2.3 is inadequate. More information is required. For example, I understand that the 4m IKONOS imagery was resampled to 2m without changing the reflectance values. What about WV-3 imagery? What resampling process did this use? Were pixel values averaged? Also, despite the authors refusal, I believe that the resampling should be added to the flowchart to make this process clear.
Answer: The pixel value of the WV-3 was resampled from nearest neighbor interpolation. We insert the pixel resampling in the flowchart.
The authors also have still not addressed the request to use PIF normalisation despite this being raised on numerous occasions.
Answer: We preferred to normalized the multispectral image bands from the Atmospheric Correction (ATCOR) available in the PCI Geomatica Software. We are sure that this is enough for our approach.
Limitations need to be added to the discussion and these should at least discuss improvements to methods which could be applied. This is important since the authors are citing the methods used in this paper as appropriate for determining land use changes in mining, however the methods used in the paper could be vastly improved.
Answer: We cited the major limitation of the GEOBIA approach in item 4.1., line 396. Furthermore, we mentioned that methodological advances should focus in the recognition and discrimination of different stages of rehabilitation in mining areas from high-resolution satellite and unmanned aerial vehicle systems,… (line 459-462).
This manuscript is a resubmission of an earlier submission. The following is a list of the peer review reports and author responses from that submission.
Round 1
Reviewer 1 Report
This is a very well constructed paper which only requires a minor check of English. I have added four suggested changes below. The research focuses on how GI Science can be used in monitoring mine rehabilitation (itself a neglected topic by many scientists, which needs more attention): the remote sensing is well explained and justified; and there is a sound discussion.
Line 95 change …mining practices, the… to …mining, the…
Line 95 change …topsoils and mining wastes… to …topsoil and mining waste…
Line 112 change Due to restricted image availability (e.g., cloud cover), high-resolution images from three different sensors (2011, 2013 Ikonos, 2012 GeoEye and 2015 WorldView-3) were acquired… to High-resolution images from three different sensors (2011, 2013 Ikonos, 2012 GeoEye and 2015 WorldView-3) were acquired and used in this research.
Line 132 change Atmospheric corrections of images were necessary… to Atmospheric correction of imagery was necessary…
Author Response
Reviewer 1: Comments and Suggestions for Authors
This is a very well constructed paper which only requires a minor check of English. I have added four suggested changes below. The research focuses on how GI Science can be used in monitoring mine rehabilitation (itself a neglected topic by many scientists, which needs more attention): the remote sensing is well explained and justified; and there is a sound discussion.
Answer: Thanks for your comments. We follow your suggestions.
Line 95 change …mining practices, the… to …mining, the…
Answer: We follow your suggestion.
Line 95 change …topsoils and mining wastes… to …topsoil and mining waste…
Answer: We follow your suggestion.
Line 112 change Due to restricted image availability (e.g., cloud cover), high-resolution images from three different sensors (2011, 2013 Ikonos, 2012 GeoEye and 2015 WorldView-3) were acquired… to High-resolution images from three different sensors (2011, 2013 Ikonos, 2012 GeoEye and 2015 WorldView-3) were acquired and used in this research.
Answer: We follow your suggestion.
Line 132 change Atmospheric corrections of images were necessary… to Atmospheric correction of imagery was necessary…
Answer: We follow your suggestion.
Reviewer 2 Report
The manuscript titled “Land cover changes in open-cast mining complexes based on high-resolution remote sensing data” analyzed the land cover changes in mining area, especially revegetation from 2011 to 2015. There are many problems in the manuscript, including lack of relevant literature, inappropriate methods, coined terms, and English writing. The manuscript needs much improvement.
Lack of literature of change detection based on high spatial resolution remote sensing data.
There are no new techniques involved. In contrast, the methods for mapping land cover and change detection have big problems.
The authors said (line 116-118) “The spatial resolution differed among the images, i.e., 116 3×3 m for Ikonos and 2×2 m for GeoEye and WorldView-3, but no resampling to the same spatial 117 scale was carried out so as to map the land cover as accurately as possible”, such statement is not true. If you don’t conduct resampling, the same feature on different resolution images have different area. In addition, resampling to the same pixel size, before classification, you don’t have to convert raster land cover map to vector, and also easily calculate areas of each class and conduct change detection based on raster.
Some terms are never heard in remote sensing field, for example, global accuracy, relative user /producer accuracy, absolute error, digital slope model. Many sentences are hard to understood and grammar errors. For example, page 3, the top paragraph.
Other comments:
What is the pixel size for LIDAR DEM and slope?
Figure 2. Flowchart. There is not logical connection between some steps. For example, NDVI and NDWI ----LiDAR terrain digital model. You should separate optical images and Lidar data, and describe the preprocessing separately.
Image segmentation: macro-segmentation and micro-segmentation: how to decide the weight for layers? Any reference or rational? I am not familiar with eCognition, but I am confused with your threshold values for classification. What is classification method used here?
In table 3, what MDT and MDD stand for?
What are complement cangas and complementary forests (forest2)? How to define them? is complement or complementary?
Change detection: raster format is easy for area calculation. The way to convert raster to vector largely impacts the shapes and areas of features. Land cover changes also are easily detected based on raster by constructing change matrix. Accuracy of land cover classification should be conducted before change detection. Only accuracies for all dates reach certain level, could change detection be conducted following.
Stratified random points means you can randomly collect certain number of points for each class one by one, it surely does not guarantee that the number of points was proportional to the size of the land cover class.
Page 8 line 199. You don’t use 2010 image.
Figure 4 is not necessary.
Figure 5: The map quality is poor; the position of coordinates is not appropriate and unit is missing; what is land cover of a couple patches on the bottom of sub-map c?
Figure 6: use lines may be more clear.
Author Response
Reviewer 2: Comments and Suggestions for Authors
The manuscript titled “Land cover changes in open-cast mining complexes based on high-resolution remote sensing data” analyzed the land cover changes in mining area, especially revegetation from 2011 to 2015. There are many problems in the manuscript, including lack of relevant literature, inappropriate methods, coined terms, and English writing. The manuscript needs much improvement.
Answer: Inappropriate methods, coined terms, and English writing were completely reviewed.
Lack of literature of change detection based on high spatial resolution remote sensing data.
Answer: A brief review was inserted in the Introduction to mention change detection techniques based on high spatial resolution remote sensing data.
There are no new techniques involved. In contrast, the methods for mapping land cover and change detection have big problems.
Answer: The methods were completely reviewed.
The authors said (line 116-118) “The spatial resolution differed among the images, i.e., 116 3×3 m for Ikonos and 2×2 m for GeoEye and WorldView-3, but no resampling to the same spatial 117 scale was carried out so as to map the land cover as accurately as possible”, such statement is not true. If you don’t conduct resampling, the same feature on different resolution images have different area.
Answer: We are sorry. We got a mistake in the text, because pixel size of Ikonos and WorldView-3 images were resampled to 2 m. Hence, all images were analyzed based on the same pixel size.
In addition, resampling to the same pixel size, before classification, you don’t have to convert raster land cover map to vector, and also easily calculate areas of each class and conduct change detection based on raster.
Answer: We know that it is possible to calculate are from raster data. However, we would need to carry out change detection analysis based on “from-to” approach. Hence, we preferred to calculate areas from individual images (2011, 2012, 2013 and 2015) and “from-to” approach (2011-2012, 2012-2013, 2013-2015 and 2011-2015) using the same method (calculate areas from polygons).
Some terms are never heard in remote sensing field, for example, global accuracy, relative user /producer accuracy, absolute error, digital slope model. Many sentences are hard to understood and grammar errors. For example, page 3, the top paragraph.
Answer: We reviewed all terms that were modified during the English review process. E.g. we changed “global accuracy” to “overall accuracy”; changed “relative user and producer accuracy” to “user’s and producer’s accuracy”; changed “absolute error” to “omission and commission errors”; and changed “digital slope model” to “slope map generated from digital terrain models”.
Other comments:
What is the pixel size for LIDAR DEM and slope?
Answer: We didn’t find this expression. We mentioned in the text that LiDAR data contains six point of elevation measurement per square meter.
Figure 2. Flowchart. There is not logical connection between some steps. For example, NDVI and NDWI ----LiDAR terrain digital model. You should separate optical images and Lidar data, and describe the preprocessing separately.
Answer: The flowchart was redesigned and optical and LiDAR data were described separately.
Image segmentation: macro-segmentation and micro-segmentation: how to decide the weight for layers? Any reference or rational?
Answer: Different weights were chosen from a trial and error (heuristics) approach to attempt to enhance specific objects that were effectively differentiated in predetermined spectral bands and indices (Baatz and Schape, 2000; Souza-Filho et al. 2018).
I am not familiar with eCognition, but I am confused with your threshold values for classification.
Answer: In the eCognition, you need to specify the threshold values for each band per class. Hence, you can use multiples bands to define one class, exactly as we defined in Table 2.
What is classification method used here?
Answer: Multiresolution segmentation classified by generating class hierarchy, which is based on fuzzy logic (Rahman and Saha,, 2008).
In table 3, what MDT and MDD stand for?
Answer: We changed MDT to DTM = digital terrain model; and changed MDD to DSM = digital slope model. These terms were inserted in the caption of the table 2.
What are complement cangas and complementary forests (forest2)? How to define them? is complement or complementary?
Answer: Objects located in the boundary between canga and forest classes (canga edge) needed to be classified as complementary canga due to they present different spectral and topographic characteristics. To identify revegetated and rehabilitated areas within mines and to separate them from misclassified forests, other classes were defined as complementary forests 1 and forests 2, that were grouped with forest class.
Change detection: raster format is easy for area calculation. The way to convert raster to vector largely impacts the shapes and areas of features. Land cover changes also are easily detected based on raster by constructing change matrix. Accuracy of land cover classification should be conducted before change detection. Only accuracies for all dates reach certain level, could change detection be conducted following.
Answer: In the GEOBIA approach, after to segment an image, it is generated segments (polygons) that represent a cluster of pixels with the same characteristics. Posteriorly, the segments are classified based on fuzzy logic. The classification product is a map represented by many polygons with defined class. This approach is different of the “pixel-to-pixel” method. As described in the manuscript (item 2.7), “In the next step, the polygons were vectorized to calculate the areas of the different classes for each year. To do so, the classifications obtained for each year were exported as shapefiles (.shp) and loaded into the ArcGIS 10.3 software (ESRI) before the area (in hectares) was calculated using the geometry calculation function present in the layer attributes table”……. “Each map was compared to its previous map by subtraction to generate a new thematic map resulting from this differentiation and a table of changes between classes”.
Stratified random points means you can randomly collect certain number of points for each class one by one, it surely does not guarantee that the number of points was proportional to the size of the land cover class.
Answer: We are sorry, In each scenes, 1024 stratified random points per classwere collected from the high-resolution images.
Page 8 line 199. You don’t use 2010 image.
Answer: Changed to 2012.
Figure 4 is not necessary.
Answer: We liked this kind of accuracy assessment presentation. It was improved by fourth reviewer. Hence, we decided to keep it in the manuscript.
Figure 5: The map quality is poor; the position of coordinates is not appropriate and unit is missing; what is land cover of a couple patches on the bottom of sub-map c?
Answer: We reorganized the figure. We didn’t know what was couple patches. Probably, they were generated during the elaboration of the figure.
Figure 6: use lines may be more clear.
Answer: We disagree. If we use lines, they cut each other become the graphic very confused.
Reviewer 3 Report
The study presents a good approach for object based segmentation and classification as an approach for land cover change detection using a collection of high spatial resolution multispectral imagery.
Overall, the paper is well articulated and effectively constructed. Hence, I would recommend this for publication.
However, some issues for consideration are as follows:
In addition to the use of UA, PA, and kappa stats of each class the authors could explore the use of F-Scores to measure inter class accuracies. In the conclusion, thoughts should be given to the operational use or transferability of this approach to other study sites or different areas of application. This would be worth including in the discussion and conclusion section of the paper.Author Response
Reviewer 3: Comments and Suggestions for Authors
The study presents a good approach for object based segmentation and classification as an approach for land cover change detection using a collection of high spatial resolution multispectral imagery.
Overall, the paper is well articulated and effectively constructed. Hence, I would recommend this for publication.
Answer: Thanks for your observations.
However, some issues for consideration are as follows:
In addition to the use of UA, PA, and kappa stats of each class the authors could explore the use of F-Scores to measure inter class accuracies.
Answer: We appreciated your suggestions. We analyzed the accuracy assessment data from Olofsson et al. (2014) as suggested by fourth reviewer.
See Olofsson, P., Foody, G. M., Herold, M., Stehman, S. V., Woodcock, C. E., & Wulder, M. A. (2014). Good practices for estimating area and assessing accuracy of land change. Remote Sensing of Environment,148, 42-57.
In the conclusion, thoughts should be given to the operational use or transferability of this approach to other study sites or different areas of application. This would be worth including in the discussion and conclusion section of the paper.
Answer: We follow your suggestion including in the discussion and conclusion the operational use or transferability of this approach to other study sites.
Reviewer 4 Report
Reviewers Comments
Remote Sensing Manuscript ID: remotesensing-579755
Type of manuscript: Article Title: Land cover changes in open-cast mining complexes based on high-resolution remote sensing data
Authors: Filipe Silveira Nascimento, Markus Gastauer, Pedro Walfir M. Souza-Filho *, Wilson R. Nascimento Jr., Diogo C. Santos, Marlene F.
Overview:
The manuscript uses a GEOBIA approach to derive changes in land cover classes relating to an iron ore mine and areas that are in the process of rehabilitation between 2011 and 2015 in the eastern Amazon. The objective of the study was to quantify the temporal and spatial changes occurring during the process of land clearing and quantify the areas undergoing progressive rehabilitation.
In general, I think that the paper is worthy of publication. However, I have some issues with the methodology that impact on the repeatability and the accuracy of the study.
The radiometric, atmospheric and georeferencing accuracy of the imagery is vague and lacking any statement of quantification of errors involved. The time series uses different sensors which is a major flaw in the study. According to the authors, this is due to cloud cover which is an acceptable justification given the location of the study. However, the issue of spatial differences was not adequately addressed. I would suggest further work to demonstrate that the decision to not resample to the limiting spatial resolution (4m) did not impact on the overall results. My concern is that the higher resolution imagery will detect different features during the segmentation process that are not picked up in the lower res imagery leading to a bias that may impact on the overall result. This is particularly an issue since the authors are quantifying precise increases/decreases to areas that have changed over time. Further, the highest resolution sensor (WorldView-3 @ 2.4m) represents the final measurement (2015) so this could be ‘finding’ more green areas compared with the coarser resolution imagery which will include more confusion between soil and veg.
The use of the Kappa statistic is not recommended. See below regarding this. Care should be taken to be clear as to what exactly is being measured. I think the authors have done a reasonable job with this. However, without ground points there is no real way of telling the quality of the revegetation and if this veg is opportunistic weeds or if the species mix is actually suitable to develop into the target ecosystems. Does the mine have monitoring records to show the rehab quality? That would be a nice addition to this study. I’m not really sure that the LiDAR added anything to the experiment? Was elevation used in the eCognition segmentation? This wasn’t clear.
Considering the points above, it would be worth the authors adding to the discussion (in further depth) the limitations of the project.
There was some awkward/difficult language that could be addressed (mostly in the results section) that I have highlighted below.
Minor things:
Line 18: I suggest that you add Brazil at the end of the sentence.
Line 19: the use of different sensors
Line 23: I would recommend against the use of the kappa statistic. Although I know it used frequently in the remote sensing literature, I do not think it is good practice. See Olofsson, P., Foody, G. M., Herold, M., Stehman, S. V., Woodcock, C. E., & Wulder, M. A. (2014). Good practices for estimating area and assessing accuracy of land change. Remote Sensing of Environment,148, 42-57. Specifically the arguments on page 51 for why this metric is not a good practice.
Line 33: Are the authors inferring this to be a global or local statement?
Line 39: references 3 & 4 –can you find local references to support this statement rather than international ones?
Lines 44-47: Can you clarify if this is a Brazilian approach or an international one?
Lines 49-52: You suggest that achieving pre-mine levels of biodiversity and function as ‘challenging’. I would suggest that full recovery of these ecosystems following mining is impossible in the short term (eg <400 years). Are there any examples in your region of either full or partial (to an acceptable level) recovery? My experience is that mines end up creating a novel ecosystem or hybrid ecosystem depending on environmental and budgetary constraints.
Lines 71: Not sure what you mean by strait benches? Maybe make it a bit more general for non-mining types to understand. ie small areas of progressive rehab associated with landform reshaping etc.
Line 75 – “rehabilitating mine lands’ do you mean areas undergoing rehabilitation?
Lines 91-94 please rewrite – difficult to understand the meaning of the sentence. Consider breaking into 2 sentences.
Lines 99-100. It is acceptable to aim for full restoration, but is it possible? Is there any evidence in this ecosystem that a full recovery is achievable? Particularly when the company is seeding a mixture of native and fast growing exotics and using fertiliser.
Line 108: Figure 1 is generally good. I am not sure what the red outline is for? Is that the study site, or does it represent a mine lease?
Line 117: WorldView-3 is 1.24x1.24m? I am concerned that not resampling to the same spatial scale may be an issue when you are trying to compare the feature detection using GEOBIA. Can the authors add further justification? The text “so as to map the land cover as accurately as possible” sounds a little vague and weak to me.
Line 120 –what methods did you use for the radiometric corrections? If you are using ground reflectance then you must have also completed atmospheric corrections? Were the images georeferenced? Figure 2 suggests that you have completed atmospheric correction and orthorectification, but there is no indication of errors associated with this. For example, if you have georeferenced imagery, you should supply the RMSE associated with the final georeferencing.
Line 126 : Table 1 consider adding into the caption that all images were captured in August (dry season). In table1 caption you correctly label ‘WorldView-3’ but in the table heading you leave out the dash (i.e WorldView 3).
Table 1 – the spatial resolution section you are using commas rather than decimal points. Eg 1,24 m needs to be 1.24 m. Same with the scene size row. Note that WorldView-3 (note the dash) is the proper spelling/punctuation for the satellite
Line 129: Figure2 The Geobia box has a misspelt word – add the ‘L’ to the word level 2. The accuracy and classification box is misleading as the authors did not conduct ‘field work’ – the use of ground control points is a desktop exercise using either API and/or knowledge of the site and was not conducted in the field. Please replace this term with another eg calibration points. If the authors actually did go into the field, this needs to be clearly stated and included in the methods.
Lines 131 – 140: I have just seen the text on atmospheric corrections – however still no data on RMSE? Consider removing the text on line 120 and adding it to section 2.3 for consistency.
Lines 141-147: please add formulas for NDVI and NDWI. Why are you using NDWI in addition to NDVI? Add some further text to justify its use. In the calculation of NDVI with WorldView-3 imagery, which NIR band did you use?
Line 147: isn’t it the other way around for NDWI? High values for NDWI = water and lower values are veg?
Table 2: Can you define what MDT and MDD are please?
Line 167: Can you please define what ‘complimentary forests’ are? Why do you have 2 levels of these forests in the micro segmentation?
Line 177: If you are using image differencing, then I would suggest that you need to use Pseudo Invariant Features (PIF) to normalise for atmospheric differences between captures.
Figure 3: I have an issue with this figure as I think it is misleading. The dotted lines suggest that the transition from reveg to Cangas and Forests is achievable but these is no evidence to support this. You have the word ‘possible’ in the caption, but I think this should be clear that it is an aspiration as these is no evidence that it is actually possible.
Line 198: If you are quoting overall results, then these should be in the main paper rather than the supplementary table.
Figure 4: Can these be colour coded?
Figure 5: This is a nice figure. Is it possible to make the forest colour slightly darker so there is more contrast between the forest and reveg? This figure has me wondering if the mine is planning to establish Canga in the same location that it was cleared?
Line 236: why is land re disturbed after it has been rehabbed?
Line 241: there is no 7d?
Line 243: there is no 7d?
Line 251: language is a bit awkward. Maybe use ‘while’ instead of ‘but’? Could the temp rehab be opportunistic veg cover and not formal rehab established through seeding and site preparation etc?
Line 261: figure 8 is good. Darker forest colour as mentioned earlier. Also check WoldView-3 (correct text)
Line 282: might be worth discussing the veg properties canopies (or saplings) may be easier to detect than smaller forbs/herbs/grasses and may have impacted on this result?
Line 297: but would need to be aligned with ground monitoring. The data could would assist this process by showing vegetation cover, but is not able to show rehabilitation success or achievement of target vegetation communities.
Lines 302-308: This section sounds a bit like mining company ‘spin’. I don’t believe that this data demonstrates that environmental liabilities are being addressed. The study shows changes in land cover classes over time and there is no qualification of rehab quality or trajectories of ecological communities towards target Cangas or Forests. Further the data shows one third of the rehab areas being re-disturbed which may indicate poor planning and a low priority for protecting rehabilitation by the mining company.
Line 316: the use of the word ‘restore’ should be cautioned. See Society for ecological restoration for the differences between the words ‘rehabilitation’ and ‘restoration’. They mean significantly different things. Restoration indicates a full recovery of the ecosystem structure and function which I would suggest is virtually impossible via your study site (in the short term), and you have provided no evidence that this is achievable.
Line 334: I’m glad you have mentioned this. However is this reference in relation to mining? Restoration when the soil profile is intact is much more achievable compared to mining rehabilitation where landforms are reshaped are topsoiled is applied to the surface (usually with stockpiled topsoil).
Lines 342-343: as above regarding achievement of legislated requirements.
Author Response
Reviewer 4: Comments and Suggestions for Authors
Reviewers Comments
Remote Sensing Manuscript ID: remotesensing-579755
Type of manuscript: Article Title: Land cover changes in open-cast mining complexes based on high-resolution remote sensing data
Authors: Filipe Silveira Nascimento, Markus Gastauer, Pedro Walfir M. Souza-Filho *, Wilson R. Nascimento Jr., Diogo C. Santos, Marlene F.
Overview:
The manuscript uses a GEOBIA approach to derive changes in land cover classes relating to an iron ore mine and areas that are in the process of rehabilitation between 2011 and 2015 in the eastern Amazon. The objective of the study was to quantify the temporal and spatial changes occurring during the process of land clearing and quantify the areas undergoing progressive rehabilitation.
In general, I think that the paper is worthy of publication. However, I have some issues with the methodology that impact on the repeatability and the accuracy of the study.
Answer: Thanks for your observation.
The radiometric, atmospheric and georeferencing accuracy of the imagery is vague and lacking any statement of quantification of errors involved.
Answer: We included some details related to radiometric, atmospheric and georeferencing accuracy of the imagery in the section 2.3. We do not consider essential to insert the reports generated during radiometric-atmospheric correction and georeferencing accuracy of the imagery, because these processes are well postulated in the literature.
The time series uses different sensors which is a major flaw in the study. According to the authors, this is due to cloud cover which is an acceptable justification given the location of the study. However, the issue of spatial differences was not adequately addressed. I would suggest further work to demonstrate that the decision to not resample to the limiting spatial resolution (4m) did not impact on the overall results. My concern is that the higher resolution imagery will detect different features during the segmentation process that are not picked up in the lower res imagery leading to a bias that may impact on the overall result.
This is particularly an issue since the authors are quantifying precise increases/decreases to areas that have changed over time. Further, the highest resolution sensor (WorldView-3 @ 2.4m) represents the final measurement (2015) so this could be ‘finding’ more green areas compared with the coarser resolution imagery which will include more confusion between soil and veg.
Answer: We explain the process in detail, showing that accuracy assessment is not so important when you are using multi-resolution and multi-date segmentation for the comparison of four single images based on objects with the same geometry, delineating spatially and spectrally consistent segments and avoiding misclassification. In other words, this approach defines segments based on all four images. The segments present the same spatial geometry and they are classified later based on spectral attributes of each image. Hence, we avoid confusion between two classes and misclassification due to spatial resolution.
The use of the Kappa statistic is not recommended. See below regarding this. Care should be taken to be clear as to what exactly is being measured. I think the authors have done a reasonable job with this. However, without ground points there is no real way of telling the quality of the revegetation and if this veg is opportunistic weeds or if the species mix is actually suitable to develop into the target ecosystems.
Answer: We agree with you. However, this manuscript does not intend to assess the quality of the revegetation or define if this vegetation is opportunistic. We intend to show that it is possible to detect and quantify the extention of mining areas and revegetation advances in the context of a open cast mine. For this approach, we add to Kappa statistic, allocation and quantity disagreement and accuracy and estimated area of land change map (Ollofson et al. 2014). In future works, we intend to assess the quality of revegetation and many ground control points must be collected to support this analysis.
Does the mine have monitoring records to show the rehab quality? That would be a nice addition to this study.
Answer: Yes. But not in sufficient amount to carry out a statistic assessment. In the future, we intend to write a manuscript about this approach.
I’m not really sure that the LiDAR added anything to the experiment? Was elevation used in the eCognition segmentation? This wasn’t clear.
Answer: A new sentence was inserted in the end of the item 2.6 to explain the use of LiDAR data, as follow: “It Is important to mention that digital terrain model (DTM) and slope maps (SM) derived from LiDAR data were important to discriminate “canga” class from mining areas”.
Considering the points above, it would be worth the authors adding to the discussion (in further depth) the limitations of the project.
Answer:A new text was inserted in the discussion section to emphasize tha main limitation of the GEOBIA approach in this study case, as follow: “The major limitation of the GEOBIA approach was determine an appropriate scale in the two levels of segmentation for better classification results. New developments must be done to try to calculate the segmentation scale based on pixel size and area of the main target to be mapped.”
There was some awkward/difficult language that could be addressed (mostly in the results section) that I have highlighted below.
Minor things:
Line 18: I suggest that you add Brazil at the end of the sentence.
Answer: We follow your suggestion.
Line 19: the use of different sensors
Answer: We follow your suggestion.
Line 23: I would recommend against the use of the kappa statistic. Although I know it used frequently in the remote sensing literature, I do not think it is good practice. See Olofsson, P., Foody, G. M., Herold, M., Stehman, S. V., Woodcock, C. E., & Wulder, M. A. (2014). Good practices for estimating area and assessing accuracy of land change. Remote Sensing of Environment,148, 42-57. Specifically the arguments on page 51 for why this metric is not a good practice.
Answer: To improve the accuracy assessment of the image classification per year, we used Pontius* et al., (2014) statistic. Furthermore, we applied Olofsson et al., (2014) for estimating area and assessing accuracy of land change as you suggested. Hence, a new item was inserted in the method section (2.9. Assessing accuracy and estimating area of land change map).
*Pontius, R. G., & Millones, M. (2011). Death to Kappa: birth of quantity disagreement and allocation disagreement for accuracy assessment. International Journal of Remote Sensing, 32(15), 4407-4429. doi:https://doi.org/10.1080/01431161.2011.552923
Line 33: Are the authors inferring this to be a global or local statement?
Answer: Depend on the point of view. The social and environmental impact of mining exploitation is local, but the iron ore production and exportation have global impact.
Line 39: references 3 & 4 –can you find local references to support this statement rather than international ones?
Answer: We follow your suggestion.
Lines 44-47: Can you clarify if this is a Brazilian approach or an international one?
Answer: We clarify this international approach.
Lines 49-52: You suggest that achieving pre-mine levels of biodiversity and function as ‘challenging’. I would suggest that full recovery of these ecosystems following mining is impossible in the short term (eg <400 years). Are there any examples in your region of either full or partial (to an acceptable level) recovery? My experience is that mines end up creating a novel ecosystem or hybrid ecosystem depending on environmental and budgetary constraints.
Answer: We consider your suggestion and changed the text as follow: “Remediation includes the revegetation, i.e., greening, and the progressive rehabilitation of biodiversity, ecosystem structure and ecosystem functioning in degraded, damaged or destroyed mine lands [2]. Although considerable progress can be achieved in short time periods [13], such mine land rehabilitation remains challenging”
Lines 71: Not sure what you mean by strait benches? Maybe make it a bit more general for non-mining types to understand. ie small areas of progressive rehab associated with landform reshaping etc.
Answer: It was wrong. We change to steep benches.
Line 75 – “rehabilitating mine lands’ do you mean areas undergoing rehabilitation?
Answer: We consider your suggestion and changed the text as follow: “The objective of this study was to provide a methodology to track land cover changes in mines, including the spatial and temporal dynamics of revegetated mine lands, with a special focus on the recognition of small revegetated areas on steep benches”.
Lines 91-94 please rewrite – difficult to understand the meaning of the sentence. Consider breaking into 2 sentences.
Answer: We rewrote the sentences.
Lines 99-100. It is acceptable to aim for full restoration, but is it possible? Is there any evidence in this ecosystem that a full recovery is achievable? Particularly when the company is seeding a mixture of native and fast growing exotics and using fertiliser.
Answer: We considered your issues and rewrote the sentence, as follow: “Furthermore, benches from permanent structures, such as waste piles, are subjected to permanent environmental rehabilitation to reduce the overall impact of mining enterprises on natural resources [14].”
Line 108: Figure 1 is generally good. I am not sure what the red outline is for? Is that the study site, or does it represent a mine lease?
Answer: Red line represents the boundaries of mining features: WP = waste pile and N = crater. We explain in the caption of figure 1.
Line 117: WorldView-3 is 1.24x1.24m?YES. I am concerned that not resampling to the same spatial scale may be an issue when you are trying to compare the feature detection using GEOBIA. Can the authors add further justification? The text “so as to map the land cover as accurately as possible” sounds a little vague and weak to me.
Answer: We resampled all satellite images to 2m in pixel size during the orthorectification process, as described in item 2.3: “The root mean square error was approximately one pixel in size. To map the land cover as accurately as possible, the pixel size of GeoEye was 2 m, whereas the Ikonos and WorldView-3 images were resampled to 2x2 m during the orthorectification process.”
Line 120 –what methods did you use for the radiometric corrections? If you are using ground reflectance then you must have also completed atmospheric corrections?
Answer: Conversions from digital number to ground reflectance data were carried out in the Atmospheric Correction - ATCOR module of the software PCI Geomatica 2016 (PCIGeomatica. Geomatica ii: Course guide. Version 0.2; PCI Geomatics: Markham, Ontario 2015; p 169).
Were the images georeferenced? Figure 2 suggests that you have completed atmospheric correction and orthorectification, but there is no indication of errors associated with this. For example, if you have georeferenced imagery, you should supply the RMSE associated with the final georeferencing.
Answer: We inserted this information in the text (item 2.3).
Line 126 : Table 1 consider adding into the caption that all images were captured in August (dry season). In table1 caption you correctly label ‘WorldView-3’ but in the table heading you leave out the dash (i.e WorldView 3).
Answer: We follow your suggestion.
Table 1 – the spatial resolution section you are using commas rather than decimal points. Eg 1,24 m needs to be 1.24 m. Same with the scene size row. Note that WorldView-3 (note the dash) is the proper spelling/punctuation for the satellite
Answer: We follow your suggestion.
Line 129: Figure2 The Geobia box has a misspelt word – add the ‘L’ to the word level 2. The accuracy and classification box is misleading as the authors did not conduct ‘field work’ – the use of ground control points is a desktop exercise using either API and/or knowledge of the site and was not conducted in the field.
Answer: We follow your suggestion and reorganized the figure.
Please replace this term with another eg calibration points. If the authors actually did go into the field, this needs to be clearly stated and included in the methods.
Answer: We follow your suggestion and mentioned this point in item 2.8. Accuracy assessment.
Lines 131 – 140: I have just seen the text on atmospheric corrections – however still no data on RMSE? Consider removing the text on line 120 and adding it to section 2.3 for consistency.
Answer: We follow your suggestion.
Lines 141-147: please add formulas for NDVI and NDWI.
Answer: We think that is not necessary to insert the formulas. We cited reference papers that describe the formulas.
Why are you using NDWI in addition to NDVI? Add some further text to justify its use. In the calculation of NDVI with WorldView-3 imagery, which NIR band did you use?
Answer: We explain why to use these two indices and mention that we use spectral bands with wavelength between 770 – 895 nm, similar with the other sensors used in this study
Line 147: isn’t it the other way around for NDWI? High values for NDWI = water and lower values are veg?
Answer: No. Higher values of NDWI enhance the water, while lower values are related to all other class, such as vegetation, soil and urban area. Hence, it is possible to recognize in detail all water bodies.
Table 2: Can you define what MDT and MDD are please?
Answer: We got a mistake. We changed MDT to DTM = digital terrain model and changed MDD to DSM = digital slope map.
Line 167: Can you please define what ‘complimentary forests’ are? Why do you have 2 levels of these forests in the micro segmentation?
Answer: We explain this in the third paragraph of the item 2.6, as follow: “Objects located in the boundary between canga and forest classes (canga edge) needed to be classified as complementary canga due to they present different spectral and topographic characteristics. Later, this class was grouped with canga class. The classification of micro-segments in the mining area class enables the differentiation between revegetated and non-revegetated mining areas (Table 3). This refinement was necessary to identify revegetated and rehabilitated areas within mines and to separate them from misclassified forests, defined as complementary forests 1 and forests 2, that were grouped with forest class. It Is important to mention that digital terrain model (DTM) and slope maps (SM) derived from LiDAR data were important to discriminate “canga” class from mining areas”.
Line 177: If you are using image differencing, then I would suggest that you need to use Pseudo Invariant Features (PIF) to normalise for atmospheric differences between captures.
Answer: The images were converted to ground reflectance to reduce the atmospheric effect. Hence, we believe that it is not necessary to use PIF to normalize atmospheric differences between captures.
Figure 3: I have an issue with this figure as I think it is misleading. The dotted lines suggest that the transition from reveg to Cangas and Forests is achievable but these is no evidence to support this. You have the word ‘possible’ in the caption, but I think this should be clear that it is an aspiration as these is no evidence that it is actually possible.
Answer: You are sure. We corrected the figure.
Line 198: If you are quoting overall results, then these should be in the main paper rather than the supplementary table.
Answer: As you recommended to carry out more robust accuracy assessment analysis, We decided to apply the determination of allocation and quantity disagreement as postulated by Pontius et al. (2014). This approach generate a large table and we decided to keep it as supplementary material. The main results are illustrated in “Figure 5”.
Figure 4: Can these be colour coded?
Answer: Unfortunately, I did not get to produce this figure in color.
Figure 5: This is a nice figure. Is it possible to make the forest colour slightly darker so there is more contrast between the forest and reveg?
Answer: We follow your suggestion.
This figure has me wondering if the mine is planning to establish Canga in the same location that it was cleared?
Answer: Exactly.
Line 236: why is land re disturbed after it has been rehabbed?
Line 241: there is no 7d?
Answer: We corrected. We changed to 7b.
Line 243: there is no 7d?
Answer: We corrected. We changed to 7b.
Line 251: language is a bit awkward. Maybe use ‘while’ instead of ‘but’?
Answer: We follow your suggestion.
Could the temp rehab be opportunistic veg cover and not formal rehab established through seeding and site preparation etc?
Answer: This is very incipient. The process is largely dominated by formal rehabilitation.
Line 261: figure 8 is good. Darker forest colour as mentioned earlier. Also check WoldView-3 (correct text).
Answer: We follow your suggestion, enhancing the figure.
Line 282: might be worth discussing the veg properties canopies (or saplings) may be easier to detect than smaller forbs/herbs/grasses and may have impacted on this result?
Answer: We believe that our approach to map revegetation inside mining areas avoid misclassification. Therefore, it does not impact our results. We discuss this aspect in the end of discussion, because this is our current research.
Line 297: but would need to be aligned with ground monitoring. The data could would assist this process by showing vegetation cover, but is not able to show rehabilitation success or achievement of target vegetation communities.
Answer: We changed the text according your suggestion.
Lines 302-308: This section sounds a bit like mining company ‘spin’. I don’t believe that this data demonstrates that environmental liabilities are being addressed. The study shows changes in land cover classes over time and there is no qualification of rehab quality or trajectories of ecological communities towards target Cangas or Forests.
Answer: We would like to say that environmental agencies (E.g. National Environment Institute) could follow the process of revegetation conducted by mining companies. We changed the text as follow: “. This technology may be helpful to monitor the revegetation of mine lands conducted by mine companies, becoming a important tool of environment monitoring by official licensing agencies.
Further the data shows one third of the rehab areas being re-disturbed which may indicate poor planning and a low priority for protecting rehabilitation by the mining company.
Answer: This is not indicating a poor planning. These mining are in expansion with many active mines in phase of iron ore exploitation. A great effort has been done to vegetate available areas for final rehabilitation.
Line 316: the use of the word ‘restore’ should be cautioned. See Society for ecological restoration for the differences between the words ‘rehabilitation’ and ‘restoration’. They mean significantly different things. Restoration indicates a full recovery of the ecosystem structure and function which I would suggest is virtually impossible via your study site (in the short term), and you have provided no evidence that this is achievable.
Answer: We agree with you. We revise al the text to do not use the term “restore”. We preferred to use “rehabilitation” or “revegetation”, depending of the context.
Line 334: I’m glad you have mentioned this. However is this reference in relation to mining? Restoration when the soil profile is intact is much more achievable compared to mining rehabilitation where landforms are reshaped are topsoiled is applied to the surface (usually with stockpiled topsoil).
Answer: We agree with you.
Lines 342-343: as above regarding achievement of legislated requirements.
Answer: We adapt this text in the manuscript.